# Connexin 43 Loss Triggers Cell Cycle Entry and Invasion in Non-Neoplastic Breast Epithelium: A Role for Noncanonical Wnt Signaling

**DOI:** 10.3390/cancers11030339

**Published:** 2019-03-08

**Authors:** Sabreen Fostok, Mirvat El-Sibai, Dana Bazzoun, Sophie Lelièvre, Rabih Talhouk

**Affiliations:** 1Department of Biology, Faculty of Arts and Sciences, American University of Beirut (AUB), Beirut 1107 2020, Lebanon; sff07@aub.edu.lb (S.F.); dbazzoun@hfcc.edu (D.B.); 2Department of Natural Sciences, School of Arts and Sciences, Lebanese American University (LAU), Beirut 1102 2801, Lebanon; mirvat.elsibai@lau.edu.lb; 3Department of Basic Medical Sciences, Purdue University, West Lafayette, IN 47907, USA; lelievre@purdue.edu; 4Purdue University Center for Cancer Research, West Lafayette, IN 47907, USA

**Keywords:** mammary gland, mammary epithelium, breast cancer, gap junctions, connexin 43, Wnt pathways, proliferation, motility, invasion, microenvironment

## Abstract

(1) Background: The expression of connexin 43 (Cx43) is disrupted in breast cancer, and re-expression of this protein in human breast cancer cell lines leads to decreased proliferation and invasiveness, suggesting a tumor suppressive role. This study aims to investigate the role of Cx43 in proliferation and invasion starting from non-neoplastic breast epithelium. (2) Methods: Nontumorigenic human mammary epithelial HMT-3522 S1 cells and Cx43 shRNA-transfected counterparts were cultured under 2-dimensional (2-D) and 3-D conditions. (3) Results: Silencing Cx43 induced mislocalization of β-catenin and Scrib from apicolateral membrane domains in glandular structures or acini formed in 3-D culture, suggesting the loss of apical polarity. Cell cycle entry and proliferation were enhanced, concomitantly with c-Myc and cyclin D1 upregulation, while no detectable activation of Wnt/β-catenin signaling was observed. Motility and invasion were also triggered and were associated with altered acinar morphology and activation of ERK1/2 and Rho GTPase signaling, which acts downstream of the noncanonical Wnt pathway. The invasion of Cx43-shRNA S1 cells was observed only under permissive stiffness of the extracellular matrix (ECM). (4) Conclusion: Our results suggest that Cx43 controls proliferation and invasion in the normal mammary epithelium in part by regulating noncanonical Wnt signaling.

## 1. Introduction

Connexins (Cxs), the building blocks of the channel-forming gap junctions (GJs), exhibit spatiotemporal expression patterns throughout development of the mammary gland [1,2,3,4,5,6]. In the human breast, Cx43 is expressed in luminal epithelial and myoepithelial compartments, while the expression of Cx26 is restricted to the luminal epithelium [7,8]. Altered expression, localization, and function of Cxs have been reported in human breast cancer tissues and in cell lines [9,10,11,12,13,14,15,16] and have been associated with developmental defects in mouse models [17,18,19], suggesting that Cxs have developmental and tumor suppressive roles. Indeed, Cx43 is proposed as an independent prognostic factor in light of its positive correlation with improved disease outcome in breast cancer patients [20,21]. Exogenous expression of Cx43 in human breast cancer cell lines reduces proliferation, invasiveness, xenograft tumor growth and metastasis and restores the differentiation capacity [22,23,24,25,26,27]. Conversely, silencing Cx43 in Hs578T human breast cancer cell line enhances proliferation and anchorage-independent growth and upregulates the expression of vascular endothelial growth factor (VEGF) [28]. Similarly, heterozygous Cx43 mutation enhances the susceptibility to lung metastasis in mice with 7,12-Dimethylbenz(a)anthracene (DMBA)-induced mammary tumors [29].

The involvement of GJ-independent mechanisms in the tumor suppressive functions of Cxs [23,24,30] suggests a cross-talk with signaling pathways that regulate tumorigenesis of the mammary gland, such as Wnt signaling [31]. The canonical or Wnt/β-catenin and the noncanonical Wnt pathways execute key roles in the development and differentiation of the mammary gland, and altered Wnt signaling is associated with breast cancer [32,33,34,35,36,37,38,39,40,41,42,43]. Although Cxs act as downstream transcriptional and functional targets of Wnt signaling in the breast tissue [44,45,46], their possible role as upstream regulators of Wnt pathways is yet to be investigated [31]. It has been proposed that reduced Cx expression and the subsequent loss of gap junctional intercellular communication (GJIC) could aid the physical detachment of cells from the tumor microenvironment and enhance their migratory capacity in the context of a primary tumor [47,48]. On the other hand, restoration of Cx expression and GJIC in advanced stages of breast cancer has been suggested to mediate the interaction of tumor cells with their microenvironment, thereby enhancing tumor metastasis [47,49,50,51]. Given that Cxs act as tumor suppressors or enhancers in a stage-specific manner [8,14,52,53,54,55,56], and considering the extracellular matrix (ECM) remodeling events in breast cancer [57,58], Cxs might act in cooperation with an altered microenvironment to induce breast tumorigenesis and invasion. The signaling pathways that act downstream of Cx loss in breast cancer and the role of the mammary epithelial microenvironment in execution of Cx signaling remain poorly investigated. In this study, we assess the activity of canonical and noncanonical Wnt pathways in Cx43-silenced non-neoplastic mammary epithelium, and we describe a role for the microenvironment as a coregulator of Cx43-driven events. 

In order to address the role of Cx43 loss in breast homeostasis, we utilized the nontumorigenic human mammary epithelial HMT-3522 S1 cell line [59]. When cultured in the presence of basement membrane components-enriched Matrigel under 3-dimensional (3-D) conditions, S1 cells assemble into growth-arrested differentiated glandular structures or acini. An apicobasal polarity axis, a central lumen, and apicolateral membrane (apical pole of the lateral membrane) expression of Cx43 that forms functional GJs characterize S1 acini (Bazzoun/Adissu et al., submitted), recapitulating the normal tissue architecture in the human breast [60,61]. Our previous studies showed that Cx43-shRNA S1 cells in 3-D culture acquire a perturbed apical polarity and mitotic spindle orientation (MSO), with loss of lumen formation (Bazzoun/Adissu et al., submitted). These observations are in line with our earlier findings that demonstrate a role for Cx43 in inducing differentiation of mouse mammary epithelial cells and reversing tumorigenesis of human breast cancer cells, via membrane GJ complex assembly mediated in part by ECM signaling and β-catenin/Cx43 association [24,62,63].

In this study, we report enhanced proliferation and cell cycle entry, upregulation of c-Myc and cyclin D1 in Cx43-shRNA S1 cells, concomitantly with the mislocalization of β-catenin and Scrib from apicolateral membrane domains in 3-D cultures. While no detectable activation of Wnt/β-catenin signaling was observed, the loss of Cx43 upregulated the expression and activity of Rho GTPases (RhoA, Rac1, and Cdc42), thereby triggering the noncanonical Wnt pathway, and enhanced ERK1/2 activity. Migratory and invasive capacity was evident only under permissive, or reduced, ECM stiffness. Our results suggest that Cx43 controls proliferation and invasion pathways in the normal mammary epithelium in part via regulating noncanonical Wnt signaling.

## 2. Results

### 2.1. Silencing Cx43 Triggers Cell Cycle Entry and Enhances the Proliferation Rate

Earlier studies demonstrated that functional GJIC and Cx43 interaction with its associated proteins observed in 3-D but not 2-D cultures of mammary epithelial cells are critical for differentiation [62,63]. We sought to investigate the effects of Cx43 silencing on the proliferation rate of nontumorigenic human mammary epithelial HMT-3522 S1 cell line [59] under 2-D and 3-D culture conditions. S1 cells form monolayers of interconnected islands with cobblestone cell morphology on plastic (2-D culture conditions), and organize into multicellular spheroidal acini, characterized by cell cycle exit, a central lumen and an apicobasal polarity axis when cultured on Matrigel, an exogenous basement membrane-rich ECM (3-D culture conditions) [60,61]. This differentiation potential is impaired in Cx43-shRNA S1 cells that exhibit perturbed apical polarity and lumen-forming ability (Bazzoun/Adissu et al., submitted). The 2-D cultures of Cx43-shRNA S1 cells revealed a significant increase in cell counts, by 50% and 33% on days 6 and 10, respectively, compared to control cells (Figure 1A). The number of dead cells was negligible and did not differ between the two groups at the different time points (data not shown). Cx43-shRNA S1 cells assembled and grew into significantly larger acini that lacked a lumen in 3-D cultures when compared to control cells (Figure 1B; left lower panel and Bazzoun/Adissu et al., submitted), with an increase in acinar area of 62%, 72%, 75%, and 82% on days 4, 6, 9, and 11, respectively (Figure 1B). In addition, silencing Cx43 significantly altered the distribution of cells in the different cell cycle phases under 2-D (Figure 2A) and 3-D culture conditions (Figure 2B) at the different time points (days 4, 6, 9, and 11 in 2-D and days 4 and 11 in 3-D). Specifically, the percentage of cells in G0/G1 phase decreased in Cx43-shRNA S1 cells compared to control cells (39–44% and 28–34% decrease in 2-D and 3-D, respectively), suggesting enhanced cell cycle entry in the cell population. This result is supported by the concomitant increase in the percentages of cells in S (up to 723% and 344% increase in 2-D and 3-D, respectively) and G2/M phases (55–119% and 79–114% increase in 2-D and 3-D, respectively). Hence, the loss of Cx43 triggers cell cycle entry and enhances the proliferation rate of S1 cells under 2-D and 3-D culture conditions. Consistent with these results, Western blotting showed that c-Myc and cyclin D1 were upregulated in Cx43-shRNA S1 cells compared to control cells under 2-D and 3-D culture conditions at the different time points (days 4, 6, 9, and 11 in 2-D and day 11 in 3-D) (Figure 2C). 

### 2.2. Silencing Cx43 Alters the Localization of Junctional and Polarity Proteins

We have previously shown that blocking Cx43-mediated GJIC in 3-D cultures of S1 cells is not sufficient to promote proliferation (Bazzoun/Adissu et al., submitted). In addition, overexpression of Cx43 in MCF-7 and MDA-MB-231 human breast cancer cells suppresses proliferation by a mechanism that does not involve GJIC [24]. Thus, we speculated the involvement of GJ-independent mechanisms in the growth-regulatory functions of Cx43. Our earlier studies in breast adenocarcinoma cell lines showed that exogenously expressed Cx43 exerts its antiproliferative effects by the assembly of GJ complexes consisting of Cx43, α-catenin, β-catenin, and ZO-2 at the membrane [24]. Coimmunoprecipitation demonstrated association of Cx43 with β-catenin and ZO-2 in control S1 cells under 2-D (Figure 3) and 3-D culture conditions (Bazzoun/Adissu et al., submitted). While the protein levels of Cx43 were markedly reduced by 90% in Cx43-shRNA S1 cells, Western blotting analysis did not show an effect for Cx43 loss on the levels of β-catenin or ZO-2 compared to control cells (Figure 4A). Similarly, immunofluorescence showed homogenous membrane distribution of β-catenin at cell–cell contacts in 2-D cultures of S1 cells and Cx43-shRNA counterparts (Figure 4B; left upper panel). Under 3-D conditions, β-catenin displayed an apicolateral membrane distribution in S1 acini (Figure 4B; left lower panel) and colocalized with Cx43 (Bazzoun/Adissu et al., submitted). Silencing Cx43 significantly altered the distribution of membranous β-catenin with 81% decrease in acini showing apicolateral localization (Figure 4B; left lower and right panels). The mislocalization of β-catenin in Cx43-shRNA S1 acini was accompanied with impaired lumen formation and acinar architecture. The levels of Scrib, a key regulator of apical polarity in epithelia, were not altered in Cx43-shRNA S1 cells compared to control cells under 2-D or 3-D culture conditions, as Western blotting analysis showed (Figure 4A). Given the asymmetric distribution of polarity complexes along the apicobasal axis of polarized epithelial cells [64], we next studied the localization of Scrib. As expected, Scrib localized at cell–cell contacts in monolayers of control and Cx43-shRNA S1 cells (Figure 4C; left upper panel). While 50% of S1 acini showed apicolateral Scrib distribution in 3-D cultures, this pattern was significantly altered in Cx43-shRNA acini (only 11% of acini expressed apicolateral Scrib), where a diffuse pattern was predominant (Figure 4C; left lower and right panels), suggesting the loss of apical polarity. Taken together, the above results indicate that silencing Cx43 alters the localization of junctional and polarity proteins in S1 cells under 3-D culture conditions, possibly through the altered assembly of GJ complexes.

### 2.3. Silencing Cx43 Activates MAPK but Not Wnt/β-Catenin Signaling

Because β-catenin was mislocalized from its apicolateral membrane domains in 3-D cultures of Cx43-shRNA S1 cells (Figure 4B; left lower and right panels), we investigated the involvement of the Wnt/β-catenin pathway in mediating proliferation downstream of Cx43 loss. Wnt/β-catenin signaling plays important roles in development and differentiation of the mammary gland and is deregulated in breast cancer [32,35,36,37,39,40]. Our earlier studies suggest negative regulation of the Wnt/β-catenin pathway downstream of Cx43 signaling as a mechanism for tumor suppression in the mammary epithelium [24]. A hallmark of Wnt/β-catenin signaling is the stabilization of cytoplasmic β-catenin, leading to its accumulation and nuclear translocation, where it mediates the expression of proliferation genes [65,66,67]. Silencing Cx43 did not alter the total levels of β-catenin in 2-D (days 4, 9, and 11) or 3-D cultures (day 11) of S1 cells, as demonstrated by Western blotting analysis (Figure 5A). However, since the majority of cellular β-catenin is associated with membrane complexes, total β-catenin levels might not accurately reflect those of cytosolic or nuclear β-catenin and are therefore not indicative of Wnt/β-catenin activation [68]. Thus, we studied the activity of GSK-3α/β, a serine/threonine protein kinase that regulates the stability of β-catenin within the β-catenin destruction complex [65,66,67]. Silencing Cx43 did not affect total GSK-3α/β or active levels (p-GSK-3α/β), as Western blotting analysis showed (Figure 5B). Furthermore, the levels of phosphorylated (p)-β-catenin (unstable isoform targeted for ubiquitination and proteasomal degradation), a readout for the kinase activity of GSK-3α/β associated with the β-catenin destruction complex and an indicator of the overall status of Wnt/β-catenin signaling, did not change in Cx43-shRNA S1 cells compared to control cells, consistent with total β-catenin levels (Figure 5B). Although this result might rule out enhanced activation of the Wnt/β-catenin pathway downstream of Cx43 silencing, we reasoned that since Cx43 associates with β-catenin at the membrane, the loss of Cx43 expression might indirectly activate Wnt/β-catenin signaling. Given that the activity of GSK-3α/β associated with the β-catenin destruction complex did not change (Figure 5B), the release of β-catenin sequestered by Cx43 at the membrane could lead to its cytoplasmic accumulation and nuclear translocation in Cx43-shRNA S1 cells. Western blotting analysis of nuclear fractions did not show an increase in β-catenin levels in Cx43-shRNA S1 cells compared to control cells (Figure 5C). Interestingly, coanalysis of T4-2 cells, the tumorigenic counterparts of S1 cells, did not show an altered level of nuclear β-catenin either (Figure 5C). These data suggest the involvement of proliferation pathways other than the Wnt/β-catenin downstream of Cx43 signaling in S1 cells. One possible candidate is the ERK1/2 pathway, since we measured alterations in the levels of c-Myc and cyclin D1 that are among the downstream targets of ERK1/2 signaling [69,70]. The ERK1/2 pathway is the best characterized among three other MAPK signaling cascades (ERK5, p38 and JNK) involved in mammary gland development and breast cancer progression [71]. While total levels of ERK1/2 remained unchanged, active levels (p-ERK1/2) were upregulated in Cx43-shRNA S1 cells compared to control cells, as Western blotting analysis showed (Figure 5D). Thus, the loss of Cx43 activates MAPK signaling, a mechanism that could induce cell cycle entry and proliferation in S1 cells. 

### 2.4. Silencing Cx43 Induces Motility and Invasion

ERK1/2 signaling is known to regulate cell motility and invasion in addition to proliferation [72,73,74]. Random motility was significantly enhanced in cultures of Cx43-shRNA S1 cells compared to control cells, as time-lapse imaging of random 2-D migration revealed (Movies 1,2). Specifically, quantitative analysis showed 41% increase in the total distance traveled by Cx43-shRNA S1 cells compared to control cells over time (Figure 6A). In transwell cell invasion assay, Cx43-shRNA S1 cells showed a significant increase of four-fold in the number of Matrigel-invading cells compared to control cells (Figure 6B; upper panel). In addition, invadopodia-like actin-rich dots were evident in 2-D cultures of Cx43-shRNA S1 cells (Figure 6B; lower panel). Both control and Cx43-shRNA S1 cells formed spheroid acini when cultured on undiluted Matrigel and did not display an apparent invasive behavior. In vitro studies revealed that ECM stiffening triggers an invasive phenotype in normal, oncogene-initiated and malignant mammary epithelial cells [75,76,77]. Compared to normal tissues, breast tumors are associated with increased ECM stiffness, which correlates with aggressive subtypes [78]. However, matrix stiffness was found to enhance matrix metalloproteinase (MMP) activity in a pancreatic cancer cell line [79], suggesting that motility and invasion of cancer cells require permissive, or reduced, ECM stiffness. This is in line with a recent demonstration of enhanced motility in breast cancer cells on matrices with moderate or low stiffness [80]. Thus, we found it reasonable to study the effects of Cx43 loss on the behavior of S1 cells in the context of variable ECM stiffness. For this purpose, we utilized 3-D cultures with different Matrigel concentrations (undiluted, 1:5, 1:10, and 1:20 dilutions). Matrix stiffness and network density are altered in response to the concentration of the Matrigel [81]. While S1 cells maintained their spheroid acinar morphology, Cx43-shRNA S1 cells displayed loss of this characteristic morphology under conditions of diluted Matrigel (1:5, 1:10 and 1:20). Indeed, Cx43-shRNA S1 acini showed progressive morphological alterations over the culture period, with an increase in size and acquisition of granular edges marked by the presence of migrating cells (Figure 6C; lower panel). We used the term “nonspheroid” to describe all observed patterns of dysmorphic structures. Such structures represented 53% and 58% of Cx43-shRNA S1 cultures on 1:5 and 1:10 diluted Matrigel, respectively, and predominated in cultures on 1:20 diluted Matrigel by day 11, forming 76% of the population of multicellular structures. The percentages of nonspheroid structures in control S1 cultures were negligible under all conditions and did not exceed 4% (Figure 6C; upper panel). These results indicate that the loss of Cx43 induces the acquisition of an invasive phenotype in S1 cells in a context-dependent manner and highlight the integration of Cx and ECM signals in determining mammary epithelial morphology and behavior. Importantly, the lumen-forming ability of control S1 cells was recapitulated under conditions of 1:5 diluted Matrigel, while a significant loss of lumen formation was noted in cultures on 1:10 and 1:20 diluted Matrigel (Figure 7A). In addition, the percentages of S1 acini with apicolateral Scrib localization did not differ between cultures on undiluted and 1:5 diluted Matrigel (Figure 7B). Since the 3-D conditions of 1:5 diluted Matrigel favored an invasive phenotype in Cx43-shRNA S1 cells while maintaining normal morphology in control cells, it becomes possible to further assess the role of Cx43 loss in invasion of S1 cells under permissive ECM stiffness.

### 2.5. Silencing Cx43 Activates Rho GTPase Signaling

Rho GTPase signaling regulates the development of the mammary gland, and its aberrant activation contributes to breast tumorigenesis [82,83]. The role of Rho GTPase signaling in motility and invasion of breast cancer cells is well established [84,85,86,87,88,89,90,91,92,93]. ERK1/2 was found to associate with and phosphorylate Rho GTPase activators (RhoGEFs) and inhibitors (RhoGAPs) to modulate their activities and enhance Rho GTPase signaling [94,95,96,97]. Thus, we next investigated the expression and activity of RhoA, Rac1, and Cdc42 as a potential downstream effect for the enhanced ERK1/2 signaling that we reported as a result of Cx43 silencing in S1 cells. To our surprise, Western blotting analysis demonstrated upregulation of all Rho GTPases in Cx43-shRNA S1 cultures under 2-D (Figure 8A) and 3-D conditions of undiluted (Figure 8B) and 1:5 diluted Matrigel (Figure 8C). Furthermore, the levels of active Rho GTPases (GTP-bound) were enhanced, as pulldown assay showed (Figure 8). These data suggest that under permissive, or reduced, ECM stiffness, the loss of Cx43 induces invasion in S1 cells by altering Rho GTPase signaling.

## 3. Discussion

In this study, we report a role for the GJ protein Cx43 in controlling proliferation and migration capabilities of the mammary epithelium. Cell proliferation and migration are fundamental for tissue development and homeostasis [98,99,100]. Deregulation of the signaling pathways that underlie these processes is associated with developmental defects and disease outcomes, such as cancer [65,101].

Interestingly, of all major mammary Cxs, Cx43 is the only isoform expressed in S1 cells, making it possible to specifically study its participation in mammary homeostasis. Our earlier findings demonstrated perturbation of apical polarity and loss of lumen formation in 3-D cultures of Cx43-shRNA S1 cells (Bazzoun/Adissu et al., submitted). S1 cells transfected with nonsilencing sequence (NSS)-shRNA phenotypically resemble control cells and sustain similar expression and localization of Cx43, ruling out possible off-target effects for the shRNA constructs. This is corroborated by the induction of apical polarization in MCF-10A acini overexpressing Cx43 at apical membrane domains, but not so in acini where Cx43 fails to localize apically (Bazzoun/Adissu et al., submitted). In this study, we show that silencing Cx43 in S1 cells triggered cell cycle entry and enhanced proliferation. In support of our findings, silencing Cx43 was shown to enhance proliferation and anchorage-independent growth in Hs578T human breast cancer cell line [28], suggesting that the decrease in Cx43 levels has a long-lasting impact throughout cancer development. The effects of silencing Cx43 on proliferation and cell cycle entry in S1 cells were reproducible under 2-D and 3-D culture conditions, suggesting that the growth-regulatory role of Cx43 is not impeded by the substratum. This finding is in line with reports demonstrating reduced proliferation of human breast cancer cell lines overexpressing Cx genes in 2-D and 3-D cultures and xenografts [22,23,24,25,30,102]. Moreover, Scrib, an apical polarity regulator localized at apicolateral membrane domains in S1 acini, was mislocalized into a diffuse pattern in Cx43-shRNA counterparts, in line with Scrib redistribution in human breast cancer tissues and cell lines [103,104]. Apicobasal polarity regulates epithelial proliferation, migration, apoptosis, morphology, and differentiation [105]. The recapitulation of the effects of Cx43 silencing on proliferation and cell cycle entry of S1 cells in 2-D cultures that lack apicobasal polarization might rule out the direct involvement of apical polarity. However, this observation does not exclude the possibility that the loss of apical polarity measured in 3-D cultures sensitizes Cx43-shRNA S1 cells to other polarity-independent mechanisms downstream of Cx43 loss. Indeed, it was previously demonstrated that the loss of apical polarity primes S1 acini into cell cycle entry, but it is not sufficient to enhance proliferation [106]. In addition, depletion of Scrib in the breast, prostate, and liver accelerates tumor progression in the presence of other tumor-driving events but is not sufficient to drive tumorigenesis [104,107,108]. Blocking GJIC in 3-D cultures of S1 cells is not sufficient to enhance proliferation, although it acts in cooperation with extrinsic proliferation signals (Bazzoun/Adissu et al., submitted). Furthermore, 2-D cultures of mammary epithelial cells have compromised GJIC [63]. This suggests that Cx43 loss induces proliferation and cell cycle entry in S1 cells by mechanisms that are independent of GJIC. Indeed, overexpression of Cx43 in human breast cancer cell lines reduces proliferation and anchorage-independent growth via a GJ-independent mechanism [24,25,27]. Our earlier findings demonstrated that this effect is partly mediated by the membrane assembly of GJ complexes that recruit β-catenin and sequester it away from the nucleus [24]. While the GJ complex (Cx43, β-catenin, and ZO-2) effectively assembled in 2-D and 3-D cultures of S1 cells, only 3-D cultures of Cx43-shRNA S1 cells exhibited evident mislocalization of β-catenin from apicolateral membrane domains, at which it associated with Cx43. In light of the dynamic cross-talk among epithelial junctions and their common associated partners [109,110], β-catenin that is not incorporated into GJ complexes possibly relocalizes to other membrane domains where it associates with different junctional complexes in Cx43-shRNA S1 cells. In endothelial cells, the altered distribution of β-catenin within the membrane is associated with impaired integrity and stability of cell–cell contacts, the presence of intercellular gaps and the enhanced endothelial permeability [111,112]. Similarly, the relocalization of membranous β-catenin in endometrial epithelial cells during the implantation window of the menstrual cycle is linked to altered intercellular adhesion necessary for trophoblast invasion [113]. Thus, the mislocalization of β-catenin downstream of Cx43 silencing in S1 cells might trigger proliferation and migration as a result of weakened cell–cell contacts. This enhanced proliferation downstream of Cx43 silencing may hence be mediated by the Wnt/β-catenin pathway, which regulates the development of the mammary gland and is deregulated in breast cancer [32,35,36,37,39,40]. Cx43 negatively regulates the Wnt/β-catenin pathway as a mechanism to induce differentiation in mammary epithelial cells or to suppress tumorigenesis in breast cancer lines [24,62]. In the present study, we did not detect any direct or indirect effects for Cx43 silencing on the activity of Wnt/β-catenin signaling, despite the upregulation of its target genes c-Myc and cyclin D1. To the best of our knowledge, this is the first demonstration of an inhibitory effect for Cxs on the expression of c-Myc and cyclin D1 in the breast tissue. This effect has been reported in osteosarcoma, glioma, hepatoma, and lung carcinoma cells overexpressing Cx genes and is associated with reduced tumorigenicity [114,115,116,117]. Interestingly, T4-2 cells, the tumorigenic counterparts of S1 cells, did not show increased Wnt/β-catenin activity compared to S1 cells, suggesting the involvement of other signaling pathways. Moreover, the activity of Wnt/β-catenin signaling is enhanced only in a subset of breast tumor tissues, and the correlation between Wnt/β-catenin activity and c-Myc or cyclin D1 expression was not consistent among reports [35,118,119,120,121]. We measured enhanced MAPK signaling, particularly ERK1/2, in Cx43-shRNA S1 cells. ERK1/2 signaling regulates mammary epithelial morphology, proliferation, motility, and invasion, among others, and is involved in mammary gland development and breast cancer progression [70,71]. In line with our findings, a link was established between Cx32 deficiency and ERK1/2 activity in liver, lung, and adrenal tumors [122,123]. Transfection of a glioma cell line with Cx30 abolishes ERK1/2 activity and thereby reduces proliferation, migration, and invasion [124]. In contrast, blocking GJs or Cx43 expression in tubular proximal epithelial cells activates ERK1/2 signaling [125]. Our results showed that silencing Cx43 in S1 cells could be associated with enhanced motility and invasion, consistent with the reduced migration, invasion, and xenograft tumor metastasis observed in breast cancer cell lines overexpressing Cx43 [24,25,26,27]. The enhanced motility and invasion were concomitant with upregulated expression and active levels of Rho GTPases (RhoA, Rac1, and Cdc42) and the formation of invadopodia-like actin-rich dots. Rho GTPases, which relay noncanonical Wnt signals, regulate growth, motility, and invasion, and are implicated in mammary gland development and breast cancer progression [42,82,83,84,86,87,126,127]. RhoA, Rac1, and Cdc42 are overexpressed in human breast tumors [33,34] and are all essential for cell migration [84,86,87,93,128], indicating that the migratory phenotype we observed in Cx43-shRNA S1 cells is triggered downstream of Rho GTPase signaling. This also accounts for the upregulation of all three Rho GTPases, given the cross-talk among them [92,93]. Consistent with our data, the knockdown of Cx43 in mouse embryonic fibroblasts leads to an increase in Rac1and RhoA activities and enhances migration [129]. Overexpression of E-cadherin in nonsmall cell lung cancer cells reduces the activity of RhoA and impairs cell migration, concomitantly with the increased membrane association of p190RhoGAP, an inhibitor of Rho GTPases [130]. In endothelial cells, the interaction of p190RhoGAP with p120-catenin, a known binding partner of E-cadherin, is required for the inhibition of RhoA [131]. A similar mechanism could be acting in S1 cells, whereby Cx43 controls Rho GTPase signaling by the membrane recruitment of RhoGAPs. Overexpression of an active form of RhoA upregulates ERK1/2 in breast cancer cells and enhances motility [132]. Alternatively, global inhibition of Rho GTPases by *Clostridium difficile* toxin B inactivates ERK1/2 signaling in neurons [133]. In contrast, ERK1/2 enhances Rho GTPase signaling through direct interaction and regulation of RhoGEF and RhoGAP activities [94,95,96,97]. Thus, ERK1/2 signaling could be activated downstream or upstream of Rho GTPases in Cx43-shRNA S1 cells, where both pathways act in synergy to induce proliferation, motility, and invasion. Importantly, migration and invasion were clearly apparent upon silencing Cx43 in 2-D cultures of S1 cells that represent nonphysiologically relevant conditions. Moreover, in 3-D cultures of S1 cells, silencing Cx43 triggered invasion only under conditions of diluted Matrigel, emphasizing the role of ECM stiffness in determining the phenotype and the invasive ability of mammary epithelial cells [75,76,77,80]. In line with our findings, a study on the effect of matrix stiffness on the behavior of mammary epithelial cells showed that stiff substrates restrict the motility of both normal and breast cancer cell lines. While the motility of breast cancer cells is enhanced on substrates with moderate or low stiffness, the migratory behavior of normal cells is less sensitive to changes in substrate stiffness [80]. Compared to normal tissues, breast cancer tissues are usually characterized by increased stiffness [78,134], but lower stiffness locally is necessary for invasion by virtue of MMP activity [79,134,135] and correlates with tumor progression and metastasis [136,137,138]. Therefore, although the invasive phenotype linked to a decrease in Cx43 expression is not observed on normal matrix stiffness mimicked by undiluted Matrigel, once cells have become cancerous and activate mechanisms to decrease matrix stiffness locally, the Cx43 loss-mediated invasive phenotype is effective. This possibility supports the overriding impact of the ECM on intracellular changes illustrated initially two decades ago [139] and explains the absence of an invasive phenotype in Cx43-shRNA S1 cells despite the enhanced Rho GTPase signaling observed on undiluted Matrigel. Interestingly, while S1 cells maintained their characteristic polarity and lumen-forming ability in 1:5 diluted Matrigel, these conditions unveiled an invasive phenotype in Cx43-shRNA S1 cells. This observation indicates that the resulting phenotype due to loss of Cx43 is partially mediated by coordinated signaling between cell–cell junctions and the mammary epithelial microenvironment, since neither the loss of Cx43 nor the compliance of the matrix was sufficient to induce invasion.

## 4. Materials and Methods

### 4.1. Cell Culture

Non-neoplastic HMT-3522 S1 human mammary epithelial cells [59], between passages 52 and 60, were routinely maintained as a monolayer on plastic (2-D culture) in chemically defined serum-free H14 medium [60,140] at 37 °C and 5% CO_2_ in a humidified incubator. H14 medium was changed every 2–3 days. HMT-3522 T4-2 cells, the tumorigenic counterparts of S1 cells, were maintained under similar conditions but on plastic coated with collagen I (BD Biosciences, Bedford, MA, USA, 354236) and in the absence of epidermal growth factor (EGF). S1 and T4-2 cells were propagated as previously described [141]. For 2-D cultures, cells were plated on plastic substrata at a density of 2.3 × 10^4^ cells/cm^2^ (S1 cells) or 1.15 × 10^4^ cells/cm^2^ (T4-2 cells). The drip method of 3-D culture was used to induce the formation of acini. Briefly, cells were plated on Matrigel^TM^ (50 μL/cm^2^; BD Biosciences, 354234) at a density of 4.2 × 10^4^ cells/cm^2^ (S1 cells) or 2.1 × 10^4^ cells/cm^2^ (T4-2 cells) in the presence of culture medium containing 5% Matrigel^TM^ [60,141]. The EGF was omitted from the culture medium after day 7 to allow completion of acinar differentiation (usually observed on day 8 or 9) [60]. For some experiments, the Matrigel^TM^ was diluted at 1:5, 1:10 or 1:20 in DMEM:F-12 (Lonza, Basel, Switzerland, BE12-719F) and allowed to solidify by incubation for 4 h at 37 °C.

### 4.2. Transfection and Infection Protocols

The recombinant retroviral constructs against Cx43 have been described previously [28]. Cx43 was downregulated in S1 cells via retroviral delivery of shRNA, as described in Bazzoun/Adissu et al. (submitted). Briefly, retroviral vectors (2 μg) containing shRNA were transfected into Phoenix packaging cells using calcium phosphate (Stratagene, La Jolla, CA, USA) according to the supplier’s protocol. For infection, filtered retroviral supernatants were applied to monolayers of S1 cells on day 3. Cells were incubated with hexadimethrine bromide (Polybrene; 6 μg/mL; Sigma, St. Louis, MO, USA) for 8 h. The infection medium was removed, and cells were incubated in regular H14 medium for 24 h. Infection was repeated two additional times, and selection with hygromycin B (150 μg/mL; Calbiochem, San Diego, CA, USA) was started 72 h after the last infection. Cx43-shRNA S1 cells were maintained, propagated, and plated similarly to S1 cells, but the H14 medium was supplemented with hygromycin B for selection. The stability of the knockdown was regularly assessed in different cell passages by Western blot analysis throughout this study.

### 4.3. Trypan Blue Exclusion Method

S1 and Cx43-shRNA S1 cells were plated in 24-well tissue culture plates (2-D). The medium was removed, and the cells were subsequently trypsinized and collected. Cells were then diluted in trypan blue at 1:1 ratio (vol/vol) and counted using a hemocytometer. The cells were counted from triplicates on days 6 and 10. To assess proliferation in 3-D cultures, cells were plated in 35-mm tissue culture plates, and acinar diameters were measured manually on days 4, 6, 9, and 11 under a phase contrast microscope. An ocular micrometer was calibrated against a stage micrometer to calculate the length of an ocular division. The number of ocular divisions spanning the acinar diameter was counted, and acinar areas were then calculated and plotted as acinar size. Fifty acini were analyzed per group.

### 4.4. Cell Cycle Analysis

S1 and Cx43-shRNA S1 cells were plated in T-25 tissue culture flasks (2-D) or 35-mm tissue culture plates (3-D). Acini were isolated from 3-D cultures as described earlier [60]. Briefly, the medium was removed and acini were released from the Matrigel^TM^ by incubation with dispase (BD Biosciences, 354235) at 37 °C. Acini were then collected by centrifugation and washed thrice with DMEM:F-12 and once with 1x phosphate-buffered saline (PBS). Cells on days 4, 6, 9, and 11 (2-D) and acini on days 4 and 11 (3-D) were trypsinized and collected by centrifugation. They were subsequently fixed using ice-cold 70% ethanol and left at −20 °C overnight. Cells were then centrifuged, and the pellet was washed twice with 1 × PBS. The pellet was resuspended in 1x PBS containing 30 μg/mL propidium iodide (Molecular Probes, Eugene, OR, USA, P3566) and RNase A. Samples were transferred to flow cytometry tubes, and 10,000 cells were analyzed per group using BD FACSAria™ III (BD Biosciences). 

### 4.5. Time-Lapse Imaging

S1 and Cx43-shRNA S1 cells were plated in 35-mm tissue culture plates (2-D) at a density of 9.2 × 10^4^ cells/cm^2^, and random motility was assessed on day 5 as previously described [142]. Images of cells were collected every 60 s for 3 h using a 20× objective. During imaging, the temperature was controlled using a Zeiss heating stage, which was set at 37 °C. The medium was buffered using HEPES and overlaid with mineral oil. The speed of cell migration was quantified using the ROI tracker plugin in the ImageJ software, written by Dr. David Entenberg. This software was used to calculate the total distance traveled by individual cells. The speed was then calculated by dividing this distance by the time (180 min) and reported in μm/min. A minimum of 15 cells were analyzed per condition. The assay was done using infinity-corrected optics on a Zeiss Observer Z1 microscope supplemented with a computer-driven Roper cooled CCD camera and operated by Zen software (Zeiss, Oberkochen, Germany).

### 4.6. Transwell Cell Invasion Assay

Six-well format cell culture inserts (8 μm pore size) were coated with 400 μL of 1:5 diluted Matrigel^TM^ and incubated at 37 °C for 4 h. 3 × 10^5^ S1 or Cx43-shRNA S1 cells were plated in the inserts in DMEM:F-12 supplemented with 1% fetal bovine serum (FBS; Sigma, F-9665). DMEM:F-12 supplemented with 10% FBS was added below the insert. Cells were incubated for 72 h and were then fixed using 4% formaldehyde in 1x PBS for 20 min at room temperature. The cells towards the inside of the insert were removed using a cotton swab, and nuclei of invading cells were stained with 1 μg/mL Hoechst 33342 (Molecular Probes, H3570) in 1 × PBS for 10 min at room temperature. The insert was then cut, mounted on a microscope slide in ProLong^®^ Gold antifade reagent (Molecular Probes, P36930), allowed to dry overnight and sealed. The inserts were examined with a fluorescence microscope, and the number of invading cells was counted and reported as fold change.

### 4.7. Immunofluorescence

S1 and Cx43-shRNA S1 cells were plated on coverslips in 12-well tissue culture plates (2-D) or 4-well chamber slides (3-D) and were stained by immunofluorescence on day 9 or 12 (2-D) or day 11 (3-D) as described earlier [60]. Briefly, cells were washed with 1 × PBS and permeabilized with 0.5% peroxide and carbonyl-free Triton X-100 in cytoskeleton buffer (100 mM NaCl, 300 mM sucrose, 10 mM PIPES, pH 6.8, 5 mM MgCl2, 1 mM pefabloc, 10 μg/mL aprotinin, 250 μM NaF). Cells were washed twice with cytoskeleton buffer and fixed in 4% formaldehyde. Cells were subsequently washed thrice with 50 mM glycine in 1x PBS and blocked. Primary antibodies used were rabbit polyclonal β-catenin (4 μg/mL; Santa Cruz Biotechnology, Dallas, TX, USA, sc-7199) and goat polyclonal Scrib (0.25 or 1 μg/mL; Santa Cruz Biotechnology, sc-11049). Secondary antibodies conjugated to Alexa Fluor 568 (red), donkey anti-rabbit (Invitrogen, Waltham, MA, USA, A-10042) and donkey anti-goat (Invitrogen, A-11057), were used at 1 μg/mL. Nuclei were counterstained with 1 μg/mL Hoechst 33342, and cells were mounted in ProLong^®^ Gold antifade reagent, allowed to dry overnight and sealed. The slides were then examined and imaged with a laser scanning confocal microscope (Zeiss, LSM710). A minimum of 50 acini were analyzed per group. For F-actin staining, S1 and Cx43-shRNA S1 cells were plated on coverslips in 12-well tissue culture plates (2-D) at a density of 9.2 × 10^4^ cells/cm^2^ and were stained on day 5. Cells were fixed using 4% formaldehyde in 1 × PBS for 10 min at 37 °C, permeabilized with 0.5% Triton X-100 in 1 × PBS for 15 min on ice and blocked with 1% bovine serum albumin (BSA) in 1x PBS for 1 h at 4 °C. Cells were subsequently stained with Rhodamine–phalloidin (1:50; Molecular Probes, R415) in 1% BSA for 30 min at room temperature and mounted in ProLong^®^ Gold antifade reagent. 

### 4.8. Preparation of Whole Cell Protein Extracts and Western Blot Analysis

S1 and Cx43-shRNA S1 cells were plated in T-75 tissue culture flasks (2-D) or 35-mm tissue culture plates (3-D). Acini were isolated from 3-D cultures as described earlier [60]. Cells were harvested from 2-D cultures by scraping in 1 × PBS and were collected by centrifugation. Cells on days 4, 6, 9, and 11 (2-D) and acini on day 11 (3-D) were lysed, and whole cell extracts were prepared in lysis buffer (2% SDS in 1 × PBS with 10 µg/mL aprotinin, 1 mM PMSF, and 250 µM sodium fluoride) [143] or in RIPA buffer (50 mM Tris HCl, 150 mM NaCl, 1% Nonidet P-40, 0.5% sodium deoxycholate, 4% protease inhibitor cocktail and 1% phosphatase inhibitor cocktail). Proteins were quantified using the DC protein assay (Bio-Rad, Hercules, CA, USA, 5000116). For Western blot analysis, equal amounts of proteins were separated on polyacrylamide gels and transferred to polyvinylidene difluoride (PVDF) membranes that were subsequently blocked with 5% milk in tris-buffered saline-Tween 20 (TBST) and immunoblotted with the following: mouse monoclonal antibodies against c-Myc (2 μg/mL; Santa Cruz Biotechnology, sc-40), cyclin D1 (2 μg/mL; Santa Cruz Biotechnology, sc-8396), Cx43 (1 or 2 μg/mL; Santa Cruz Biotechnology, sc-271837, and 5 μg/mL; Invitrogen, 13-8300), GSK-3α/β (1 μg/mL; Santa Cruz Biotechnology, sc-7291), p-GSK-3α/β (0.5 μg/mL; Santa Cruz Biotechnology, sc-81496), p-β-catenin (2 μg/mL; Santa Cruz Biotechnology, sc-57535), ERK1/2 (1 μg/mL; Santa Cruz Biotechnology, sc-514302), and p-ERK1/2 (4 μg/mL; Santa Cruz Biotechnology, sc-7383), goat polyclonal antibodies against Scrib (1 μg/mL; Santa Cruz Biotechnology, sc-11049) and ZO-2 (2 or 4 μg/mL; Santa Cruz Biotechnology, sc-8148) and rabbit polyclonal antibodies against β-catenin (0.04 or 0.2 μg/mL; Santa Cruz Biotechnology, sc-7199) and Cx43 (2.5 μg/mL; Invitrogen, 71-0700). Secondary antibodies conjugated to HRP, goat anti-rabbit (Abcam, Cambridge, UK, ab6721), goat anti-mouse (Abcam, ab6789), and rabbit anti-goat (Abcam, ab6741), were used at 0.13 μg/mL. Equal protein loading was verified by immunoblotting for lamin B (rabbit polyclonal, 0.05 μg/mL; Abcam, ab16048) or GAPDH (mouse monoclonal, 0.4 or 1 μg/mL; Santa Cruz Biotechnology, sc-47724). Protein levels were quantified using ImageJ software and normalized to lamin B or GAPDH.

### 4.9. Coimmunoprecipitation

S1 and Cx43-shRNA S1 cells were plated in T-75 tissue culture flasks (2-D). Cells on day 6 were harvested by scraping in 1 × PBS, collected by centrifugation and subjected to coimmunoprecipitation as previously described [62] with the following modifications: whole cell lysates were sheared using a 27-gauge needle, protein G sepharose beads (GE Healthcare, Chicago, IL, USA, 17-0618-01) were used for 300 μg of proteins with 0.25 μg rabbit polyclonal Cx43 antibody (Invitrogen, 71-0700), and samples were resolved on 12% polyacrylamide gels.

### 4.10. Pulldown Assay

S1 and Cx43-shRNA S1 cells were plated in T-75 tissue culture flasks (2-D) or 35-mm tissue culture plates (3-D). Acini were isolated from 3-D cultures as previously described [60]. The pulldown assays were performed using the RhoA/Rac1/Cdc42 Activation Assay Combo Kit (Cell Biolabs, San Diego, CA, USA, STA-405) following the manufacturer’s instructions. Briefly, lysates of cells on day 9 (2-D) and acini on day 11 (3-D) were incubated with PAK PBD agarose beads (for Rac-GTP and Cdc42-GTP pulldown) for 1 h at 4 °C with gentle agitation. The samples were then centrifuged, and the pellets were washed several times. Subsequently, the pellets were resuspended in Laemmli sample buffer and boiled for 5 min. GTP-Rac1 and GTP-Cdc42 were detected by Western blotting using mouse monoclonal antibodies against Rac1 or Cdc42, respectively. Lysates were collected prior to the incubation with PAK PBD agarose beads and were immunoblotted with mouse monoclonal antibodies against RhoA, Rac1, and Cdc42 for detection of total GTPase levels. 

### 4.11. Subcellular Fractionation

S1, Cx43-shRNA S1, and T4-2 cells were plated in 35-mm tissue culture plates (3-D). Acini and tumor nodules were isolated from 3-D cultures as previously described [60], and subcellular fractionation was performed using the Qproteome Cell Compartment Kit (Qiagen, Hilden, Germany, 37502) following the manufacturer’s instructions. The nuclear fractions of acini on day 7 (T4-2) or day 11 (S1 and Cx43-shRNA S1) were quantified and analyzed by Western blotting using rabbit polyclonal β-catenin antibody (0.2 μg/mL; Santa Cruz Biotechnology, sc-7199). Equal protein loading was verified by immunoblotting for lamin B (rabbit polyclonal, 0.2 μg/mL; Abcam, ab16048). The purity of nuclear fractions was assessed by immunoblotting for GAPDH (mouse monoclonal, 0.4 μg/mL; Santa Cruz Biotechnology, sc-47724) and Tim23 (mouse monoclonal, 1 μg/mL; Santa Cruz Biotechnology, sc-514463), a membrane marker.

### 4.12. Statistical Analysis

Data are presented as means ±S.D. Statistical comparisons were done using Microsoft Excel 2010 software. Unpaired *t*-test was used for comparison of two groups. *p* < 0.05 was considered significant.

## 5. Conclusions

Our results demonstrate that the loss of Cx43 induces the noncanonical Wnt pathway, which acts through Rho GTPase signaling to trigger cell cycle entry, motility, and invasion in the phenotypically normal mammary epithelium. We propose that while the loss of Cx43 expression contributes to breast cancer initiation by perturbing apical polarity and normal morphology (Bazzoun/Adissu et al., submitted), it activates invasion pathways that become effective under permissive mechano-transducing cues from the matrix, suggesting a long-lasting impact for Cx43 loss on breast cancer development.

## Figures and Tables

**Figure 1 cancers-11-00339-f001:**
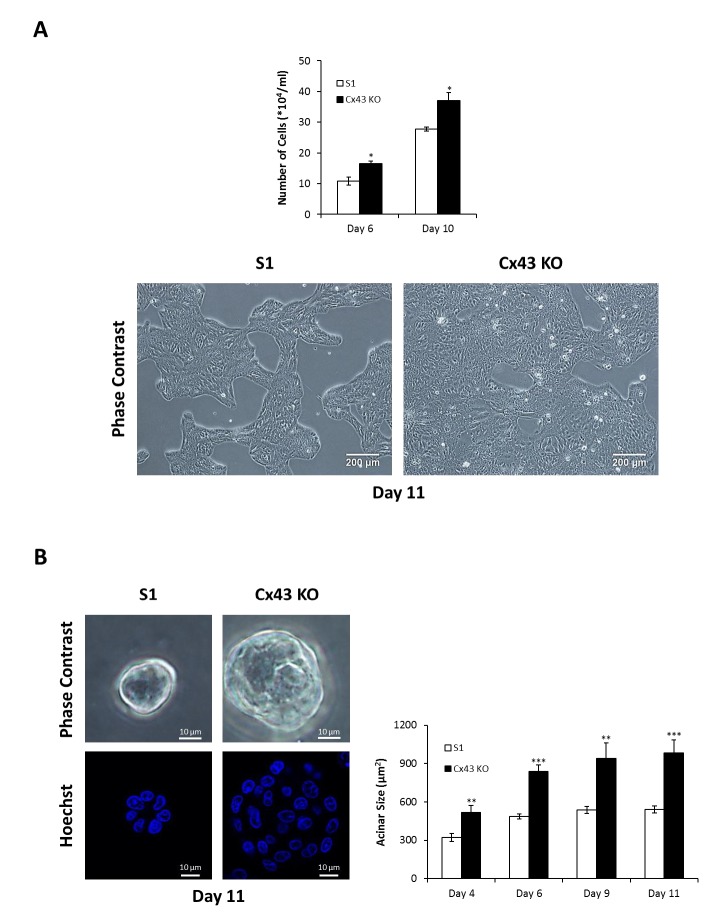
Silencing Cx43 enhances the proliferation rate of S1 cells under 2-D and 3-D culture conditions. S1 and Cx43-shRNA S1 cells (Cx43 KO) were cultured under 2-D (**A**) or 3-D conditions (**B**). Proliferation rate was assessed by cell counting on days 6 and 10 in 2-D (**A**; upper panel) and by measurement of acinar diameter on days 4, 6, 9, and 11 in 3-D (**B**; right panel). An ocular micrometer calibrated against a stage micrometer was used, and acinar areas were then calculated and plotted as acinar size. Fifty acini were analyzed per group. The values depicted in histograms are the means (±S.D.) of cell counts (**A**; upper panel) or acinar sizes (**B**; right panel) from three independent experiments. Unpaired *t*-test; * *p* < 0.05, ** *p* < 0.01, *** *p* < 0.001. Representative images of cells on day 11 in 2-D (**A**; lower panel) and in 3-D (**B**; left panel) are shown. Nuclei were stained with Hoechst (blue; **B**; left lower panel).

**Figure 2 cancers-11-00339-f002:**
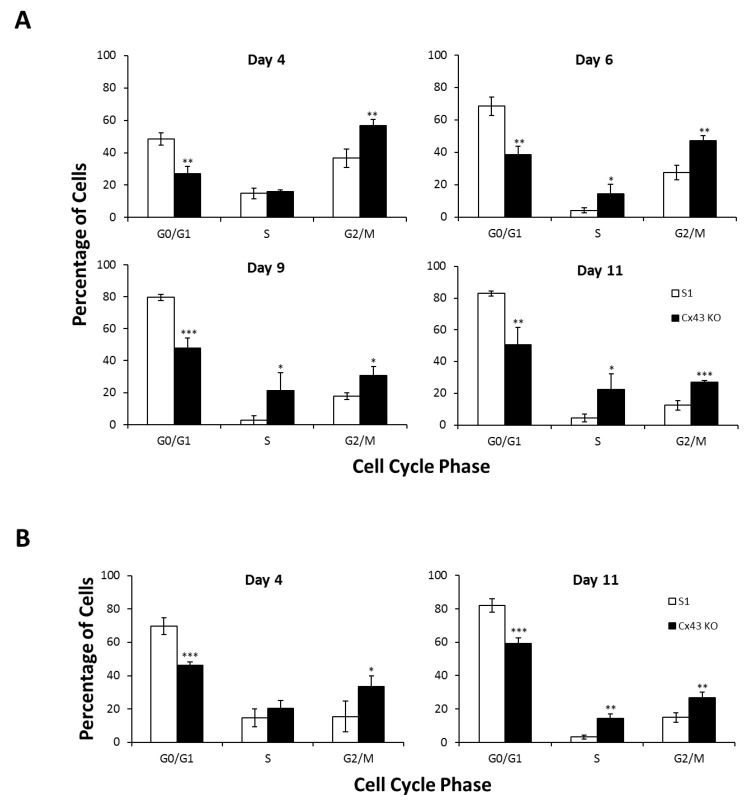
Silencing Cx43 triggers cell cycle entry and upregulates the expression of cell cycle genes in S1 cells under 2-D and 3-D culture conditions. S1 and Cx43-shRNA S1 cells (Cx43 KO) were cultured under 2-D (**A**,**C**; left panel) or 3-D conditions (**B**,**C**; right panel). A and B. Cell cycle analysis was performed by flow cytometry on days 4, 6, 9, and 11 in 2-D (**A**) and on days 4 and 11 in 3-D (**B**). The values depicted in histograms are the means (±S.D.) of cell percentages in the different cell cycle phases from three independent experiments. Unpaired *t*-test; * *p* < 0.05, ** *p* < 0.01, *** *p* < 0.001. (**C**) Total proteins were extracted on days 4, 6, 9, and 11 in 2-D (left panel) and on day 11 in 3-D (right panel). Expression of c-Myc and cyclin D1 was assessed by Western blotting. Lamin B served as loading control. The values depicted in the histogram (right lower panel) are the means of fold change in c-Myc or cyclin D1 expression in 3-D normalized to that of Lamin B from three independent experiments. Fold change in normalized expression is set to 1 in S1 cells.

**Figure 3 cancers-11-00339-f003:**
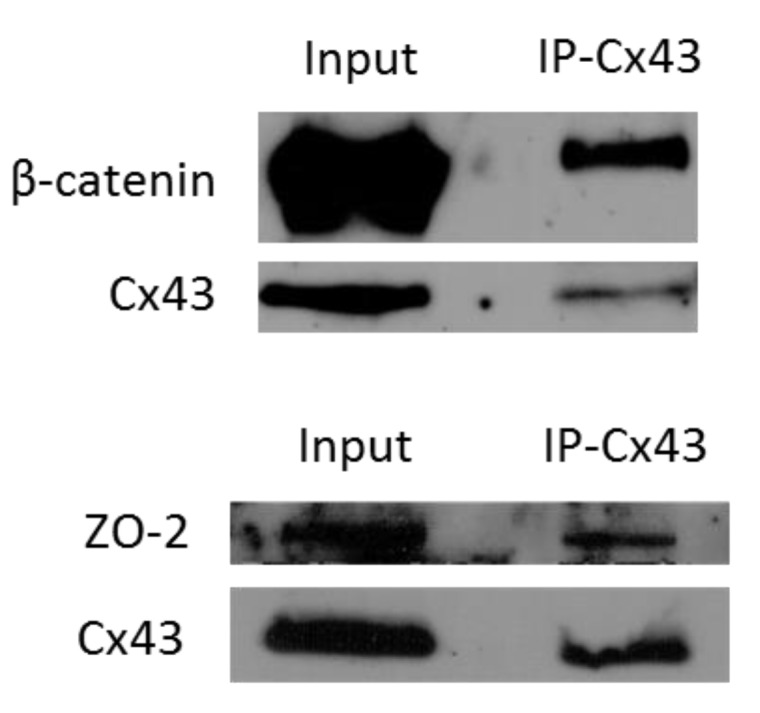
Cx43 assembles GJ complexes in S1 cells under 2-D culture conditions. S1 cells were cultured under 2-D conditions. Total proteins were extracted on day 6. Association of Cx43 and β-catenin (upper panel) or ZO-2 (lower panel) was assessed by coimmunoprecipitation (IP) of Cx43 followed by Western blotting for detection of Cx43, β-catenin and ZO-2. The input served as a control. Three independent experiments were performed for Cx43-β-catenin association and two for Cx43-ZO-2 association.

**Figure 4 cancers-11-00339-f004:**
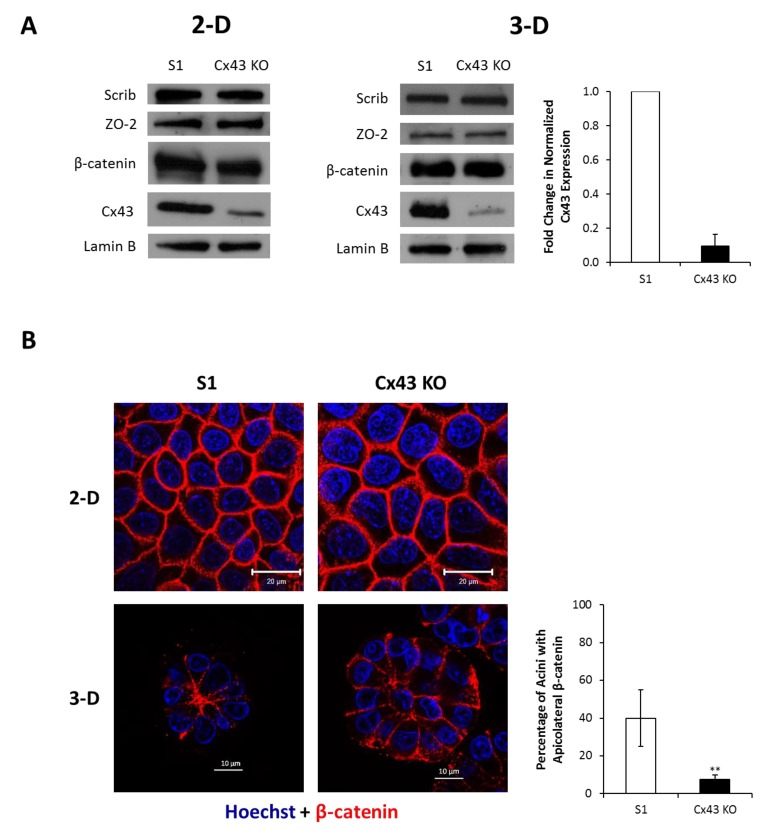
Silencing Cx43 alters the localization of junctional and polarity proteins in S1 cells under 3-D culture conditions without affecting their expression levels. S1 and Cx43-shRNA S1 cells (Cx43 KO) were cultured under 2-D or 3-D conditions. (**A**) Total proteins were extracted on day 9 in 2-D (left panel) and on day 11 in 3-D (right panel). Expression of Scrib, ZO-2, β-catenin, and Cx43 was assessed by Western blotting. Lamin B served as loading control. Three independent experiments were performed. The values depicted in the histogram (right panel) are the means of fold change in Cx43 expression in 3-D normalized to that of Lamin B from three independent experiments. Fold change in normalized expression is set to 1 in S1 cells. (**B**) Localization of β-catenin (red) was assessed by immunofluorescence on day 9 in 2-D (left upper panel) and on day 11 in 3-D (left lower panel). Nuclei were counterstained with Hoechst (blue). The values depicted in the histogram are the means (±S.D.) of acini percentages with apicolateral β-catenin from three independent experiments. One hundred acini were analyzed per group. Unpaired *t*-test; ** *p* < 0.01. (**C**) Localization of Scrib (red) was assessed by immunofluorescence on day 12 in 2-D (left upper panel) and on day 11 in 3-D (left lower panel). Nuclei were counterstained with Hoechst (blue). The values depicted in the histogram are the means (±S.D.) of acini percentages with apicolateral Scrib from three independent experiments. One hundred acini were analyzed per group. Unpaired *t*-test; *** *p* < 0.001.

**Figure 5 cancers-11-00339-f005:**
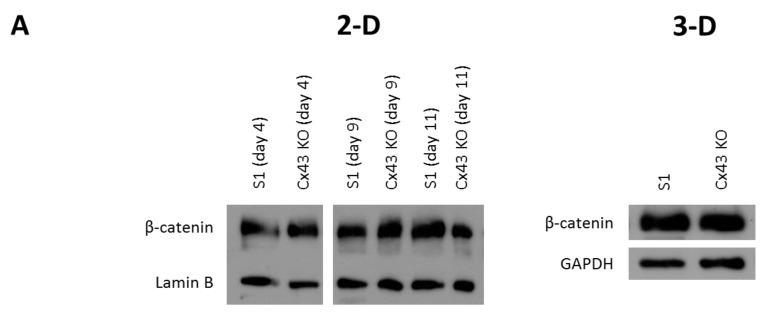
Silencing Cx43 activates MAPK but not Wnt/β-catenin signaling in S1 cells under 3-D culture conditions. S1 and Cx43-shRNA S1 cells (Cx43 KO) were cultured under 2-D (**A**) or 3-D conditions (**A**–**D**). (**A**,**B**,**D**) Total proteins were extracted on days 4, 9, and 11 in 2-D (**A**; left panel) and on day 11 in 3-D (**A**; right panel, **B**,**D**). Expression of β-catenin, p-β-catenin, GSK-3α/β, p-GSK-3α/β, ERK1/2, and p-ERK1/2 was assessed by Western blotting. Lamin B and GAPDH served as loading controls. Three independent experiments were performed. The values depicted in the histogram (**D**; right panel) are the means of fold change in p-ERK1/2 levels in 3-D normalized to that of GAPDH from three independent experiments. Fold change in normalized levels is set to 1 in S1 cells. (**C**) Nuclear proteins were extracted on day 11 (S1 and Cx43 KO cells) or on day 7 in 3-D (T4-2 cells). β-catenin levels were assessed by Western blotting. Lamin B served as loading control. Purity of nuclear fractions was determined by analyzing cytosolic (GAPDH) and membrane markers (Tim23). Total proteins extracted on day 11 in 3-D (S1 cells) were co-analyzed as a control for the detection of GAPDH and Tim23. Three independent experiments were performed.

**Figure 6 cancers-11-00339-f006:**
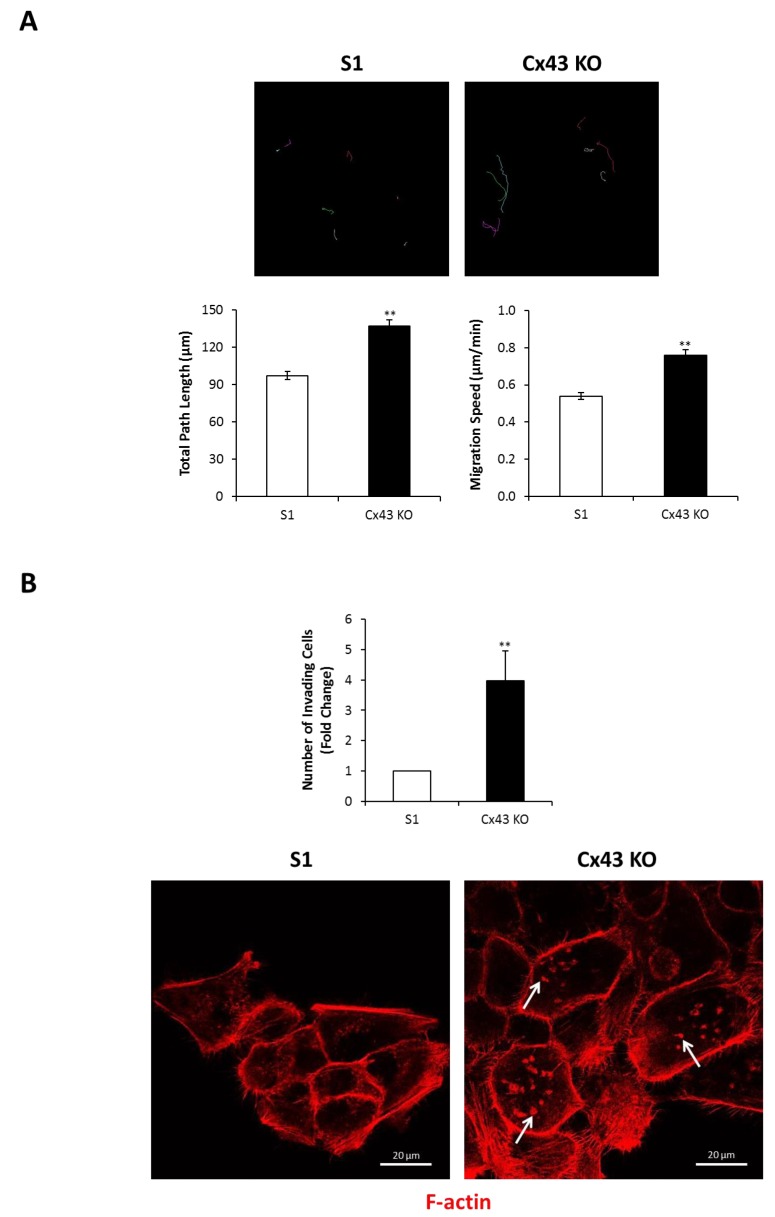
Silencing Cx43 induces motility and invasion in S1 cells. (A) S1 and Cx43-shRNA S1 cells (Cx43 KO) were cultured under 2-D conditions. Motility was assessed by time-lapse imaging on day 5. The total paths of representative S1 and Cx43 KO cells from time-lapse movies (upper panel) are shown (different colors represent different cells). Histograms show the quantification of cell motility. A total of 50 cells were analyzed per group. The values depicted are the means (±S.D.) of total path lengths (left lower panel) or migration speeds (right lower panel) from two independent experiments. Unpaired *t*-test; ** *p* < 0.01. (B) Invasion of S1 and Cx43 KO cells across diluted Matrigel (1:5) was assessed by transwell cell invasion assay (upper panel). The values depicted in the histogram are the means (±S.D.) of fold change in number of Matrigel-invading cells from three independent experiments. Unpaired *t*-test; ** *p* < 0.01. Representative images of cells cultured under 2-D conditions and stained for F-actin on day 5 (lower panel) are shown. Arrows indicate invadopodia-like actin-rich dots. (C) S1 and Cx43 KO cells were cultured under 3-D conditions atop of different Matrigel dilutions (undiluted, 1:5, 1:10 and 1:20 dilutions). Invasion was assessed by counting nonspheroid structures on day 11. One hundred structures were analyzed per group. The values depicted in the histogram (upper panel) are the means (±S.D.) of nonspheroid structures from three independent experiments. Unpaired *t*-test; *** *p* < 0.001. Representative images of cells (lower panel) are shown. Arrows indicate migrating Cx43 KO cells. Undil; Undiluted.

**Figure 7 cancers-11-00339-f007:**
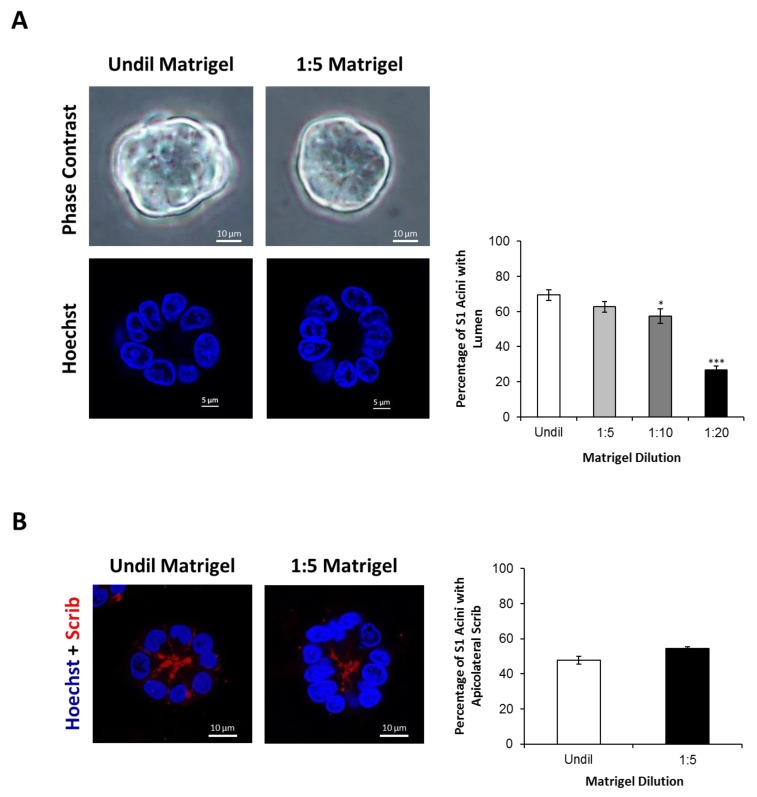
S1 cells maintain their characteristic spheroid morphology, lumen-forming ability, and polarity under 3-D culture conditions of reduced ECM stiffness. S1 cells were cultured under 3-D conditions atop of different Matrigel dilutions (undiluted, 1:5, 1:10 and 1:20 dilutions). Lumen formation (**A**) and polarity (**B**), demonstrated by apicolateral Scrib localization (red; **B**; left panel), were assessed by immunofluorescence on day 11. (**A**) Fifty acini were analyzed per condition. The values depicted in the histogram (right panel) are the means (±S.D.) of acini percentages with lumen from three independent experiments. Unpaired *t*-test; * *p* < 0.05, *** *p* < 0.001. Representative images of acini (left panel) are shown. Nuclei were stained with Hoechst (blue; left lower panel). (**B**) Sixty-seven acini were analyzed per condition. The values depicted in the histogram (right panel) are the means (±S.D.) of acini percentages with apicolateral Scrib from two independent experiments. Unpaired *t*-test. Representative images of acini (left panel) are shown. Nuclei were counterstained with Hoechst (blue). Undil; Undiluted.

**Figure 8 cancers-11-00339-f008:**
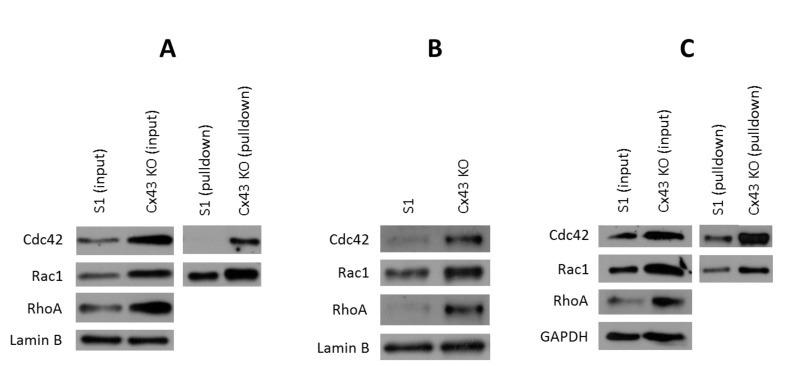
Silencing Cx43 upregulates the expression and activity of Rho GTPases in S1 cells under 2-D and 3-D culture conditions. S1 and Cx43-shRNA S1 cells (Cx43 KO) were cultured under 2-D (**A**), 3-D conditions of undiluted (**B**) or diluted Matrigel (1:5; **C**). Total proteins were extracted on day 9 in 2-D (**A**) and on day 11 in 3-D (**B**,**C**). Expression of RhoA, Cdc42, and Rac1 was assessed by Western blotting. Lamin B and GAPDH served as loading controls. Three independent experiments were performed. Activity of Rho GTPases was assessed by pulldown assay followed by Western blotting for detection of active Rho GTPases. Three independent experiments were performed.

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
