# Peer review of "Connexin 43 Loss Triggers Cell Cycle Entry and Invasion in Non-Neoplastic Breast Epithelium: A Role for Noncanonical Wnt Signaling"

_cancers, 2019, doi:10.3390/cancers11030339_

Round 1
Reviewer 1 Report
The notion that intercellular signaling via membrane channels of gap junctions, commonly referred to as gap junctional intercellular communication, may also regulate neoplasia by inhibiting the proliferation of cancer cells and controlling differentiation and cell polarity is intriguing. The family of constituent proteins of gap junctions, called connexins, is large, and several lines of evidence now point to the potential tumor suppressive role of these proteins. Because some connexins are expressed in a tissue-specific, cell-cycle-specific, and differentiation-specific manner and may be altered and regulated distinctly in different tissues, as well as depending on the genetic background, elegant three-dimensional systems are direly needed to determine and understand the mechanism by which cell-cell communication alters or modulates cell cycle, differentiation, tumor initiation, and progression.
Connexin43 is expressed ubiquitously in most tissues. Several studies have demonstrated both tumor suppressing and enhancing role of Cx43 when expressed in established cell lines, which are cultured under two-dimensional (2-D) culture conditions. However, 2-D cultures do not mimic the in vivo environment and have major limitations. Several studies have reported both tumor enhancing and suppressive effects of Cx43 expression in many cell lines including breast cancer cell lines. Moreover, the expression of Cx43 is turned on in tumors. Earlier studies from Dr. Laird’s laboratory showed that the expression of Cx43 was absent in several established cell lines and that forced expression of Cx43 suppressed their malignant behavior both in 2-D and three-dimensional (3-D) cultures. In this study, the authors have reevaluated the role of Cx43 in regulating cell growth and differentiation/cell polarity in 3-D cultures using a non-transformed breast cell line, HMT-3522 S1, that undergoes acinar morphogenesis and differentiation in 3-D cultures. This cell line, which was developed in Dr. Bissell’s laboratory, is thus useful in studying the role of Cx expression in regulating cell growth and differentiation or the polarized phenotype of cells. The major findings reported in this manuscript are summarized as follows:
1. HMT-3522 S1 cells express Cx43 and undergo acinar morphogenesis in 3-D cultures whereas S1 cells from which Cx43 has been knocked down fail to undergo acinar morphogenesis. This loss is accompanied by loss of apicolateral polarity in 3-D.
2. Knock-down of Cx43 causes mislocalization of β-catenin and Scrib from the apicolateral membrane domains.
3. Knockdown of Cx43 accelerated cell cycle entry, enhanced proliferation, and increased the cyclin D1 and c-Myc expression levels in 2-D and 3-D cultures without activating canonical Wnt/β-catenin signaling pathway.
4. Knockdown of Cx43 also activated ERK1/2 pathway as well as Rho GTPase signaling, a downstream effector of non-canonical Wnt/β-catenin signaling pathway.
5. Knockdown of Cx43 altered invasion only under conditions where stiffness of the extracellular matrix was reduced.
Based on these salient findings the authors concluded that Cx43 expression regulates the proliferation, invasion, and apicolateral polarity by altering non-canonical Wnt/β-catenin signaling pathway.
The major strength of this study is the choice of a well-characterized cell line that undergoes acinar morphogenesis, and which is well accepted to be an in vivo correlate of the normal human mammary epithelial cells. The authors have published several interesting papers related to the role of Cxs in regulating the behavior of mammary epithelial cells.
A major weakness of the proposed studies is the lack of link between alteration in cell proliferation, loss of apicolateral polarity, upregulation of cyclin D1 and c-Myc, activation of non-canonical Wnt/β-catenin signaling pathway and alteration in invasion under specific conditions. It is unclear to this reviewer as to how would knockdown of Cx43 induce all these changes! This has not been made clear in the discussion. In addition, there is no other control for the knockdown. The transformed counterpart of S1 cells, HMT-3522 T4-2 cells, were used only in one experiment. Another major pitfall of this study is referral to the data from a manuscript that has been submitted for publication and from which no Figures have been included for the reviewer to have a look. It appears to this reviewer that these data are crucial for the appreciation of this important study. The discussion is very long and superficial with no relevance to the key data whose possible interpretation is being discussed.
The specific comments are as follows:
Introduction:
Line 52: The word “gap junctions” should replace the word channel, as it has not been mentioned that gap junctions are formed of cell-cell channels.
Lines 56-57: It is a paradoxical statement. This sentence clearly shows that Cx expression affects Wnt signaling. So this sentence should be modified.
Lines 58-61: This is not a correct statement. Dissemination is linked to loss of cadherin expression but not to Cx expression. Please site the relevant reference or modify this statement.
Results:
Lines 101-102: These are crucial data and should have been included in this manuscript.
Figure 1 B: Please describe clearly in the method how acinar sizes were measured and how many acini were scored? Measuring areas could be misleading when measuring proliferation in 3-D cultures.
Figure 2: Expression level of Cx43 in S1 and knocked-down cells should by shown? Also, please describe the specifics of ShRNAs and the details of the vector used in “Methods”. How many sequences were tried and how stable was the knockdown?
Lines 146-148: This statement cannot be made with regard to work that is barely submitted. Please replace channel-independent with "gap junction-independent”.
Lines 157-159: These data should be shown or this statement removed.
Figure 3: This Figure and legend is confusing. Also the Figure is split. In addition, the legend is hard to comprehend with regard to reference points in the Figure. Figure 3 legend does not describe Figure 3B. In addition, in 3-D cultures, β-catenin appears at the apical domain, which is unlikely. It should be on the lateral domain. Markers specific for the apical or basolateral domain would improve this Figure. Upper immunofluorescence panels show cells under different magnification or else knockdown of Cx43 increases cell size as well. Please clarify.
Figure 4 legend line 195: Between 100-XXX acini were scored per group is a better statement.
Line 200: It is not clear from Figure 4 B, what has happened to the localization of β-catenin. In addition, the use of the word apicolateral domain is confusing. It should be clarified what the authors mean by apicolateral domain!
Lines 220-222: Colocalization of Cx43 and β-catenin should be shown. This will help the readers appreciate the colocalization status of Cx43 and beta-catenin.
Lines 264-266: It is not clear what the authors mean by the "phenotype".
Lines 266-268: This is a vague statement. Do the authors mean that the ECM stiffness increases invasiveness of breast tumors? Alternatively, do they mean something else? Please clarify.
Line 271: The term "permissive ECM stiffness" should be clarified.
Figure 6 B: There is a huge cell size difference between control and Cx43 knockdown cells in immunofluorescence images. It appears that we are dealing here with multiple population of cells.
Figure 7 legend: Apicolateral is a confusing term. Is Scrib localized apically or basolaterally or at the junction of apical and lateral domains? It should be clarified.
Line 330 and paragraph: What is the driving force here? Is it activation of ERK1/2 pathway or Rho GTPase signaling? Is there a potential link between these pathways? A clear rationale would help the readers.
Figure 8: It looks like that all GTPases (Rho, Rac and Cdc42) are activated! Different GTPases do different things. A brief description or explanation would help the readers and would clarify what the authors are trying to emphasize specifically?
Lines 349-370: These lines have nothing to do with the discussion. Please discuss the data and do not review the literature here. This is basically a repetition of the "Introduction" to some extent.
Lines 372-373: Once again this is a very distracting statement. These data should be available if this reviewer is to provide constructive critique to the authors.
Lines 394-395: Again these data should be made available. Otherwise, it is not possible to appreciate the significance of this study.
Lines 400-403: This statement is not justified by the data shown in this manuscript. It should be modified or deleted.
Lines 418-427: These lines center on β-catenin with no reference to Cx43. These lines are diffuse and are not related to the main findings of the manuscript.
Lines 435-437: Please discuss the significance of upregulation of c-Myc and cyclin D1 with regard to Cx43 knockdown. This is potentially an important finding.
METHODS
Lines 498-499: The details of which sequence was used to knock down Cx43 should be described. This key methodology is once again referred to as manuscript submitted is also discomforting.
Lines 506-508: It is not clear how exactly the acinar areas were measured and how many acini were scored.
Lines 520-521: Approximately how many acini were used and how many cells from these acini were subjected to cell cycle analysis by BD FACSArea III.
Lines 593-598: How much total protein was used for the co-immunoprecipitation experiments.
This reviewer's comments to the manuscript written by Fostok et al. with regard to role of Cx43 in non-neoplastic breast epithelium were intended to help the authors. The authors have made substantial contributions with regard to the role of Cxs in regulating breast cancer cell proliferation and differentiation. However, an important set of references were omitted. Specifically, the authors failed to cite several important seminal findings reported by James Trosko's group (with CC Chang and others) which described the role of gap junctions in regulating the malignant phenotype, differentiation, and stem cell like characteristics of breast cancer epithelial cells both in 2-D and 3-D organotypic cultures. This reviewer thus encourages the authors to cite some of those references.
I hope the authors would find these comments useful. Particularly, the discussion is diffuse and should be reduced to discussion of key points.
Author Response
Please note that referees’ comments appear below in bold. Rebut by the author appears in regular non-bold font.
Thank you
Comments and Suggestions for Authors
The notion that intercellular signaling via membrane channels of gap junctions, commonly referred to as gap junctional intercellular communication, may also regulate neoplasia by inhibiting the proliferation of cancer cells and controlling differentiation and cell polarity is intriguing. The family of constituent proteins of gap junctions, called connexins, is large, and several lines of evidence now point to the potential tumor suppressive role of these proteins. Because some connexins are expressed in a tissue-specific, cell-cycle-specific, and differentiation-specific manner and may be altered and regulated distinctly in different tissues, as well as depending on the genetic background, elegant three-dimensional systems are direly needed to determine and understand the mechanism by which cell-cell communication alters or modulates cell cycle, differentiation, tumor initiation, and progression.
Connexin43 is expressed ubiquitously in most tissues. Several studies have demonstrated both tumor suppressing and enhancing role of Cx43 when expressed in established cell lines, which are cultured under two-dimensional (2-D) culture conditions. However, 2-D cultures do not mimic the in vivo environment and have major limitations. Several studies have reported both tumor enhancing and suppressive effects of Cx43 expression in many cell lines including breast cancer cell lines. Moreover, the expression of Cx43 is turned on in tumors. Earlier studies from Dr. Laird’s laboratory showed that the expression of Cx43 was absent in several established cell lines and that forced expression of Cx43 suppressed their malignant behavior both in 2-D and three-dimensional (3-D) cultures. In this study, the authors have reevaluated the role of Cx43 in regulating cell growth and differentiation/cell polarity in 3-D cultures using a non-transformed breast cell line, HMT-3522 S1, that undergoes acinar morphogenesis and differentiation in 3-D cultures. This cell line, which was developed in Dr. Bissell’s laboratory, is thus useful in studying the role of Cx expression in regulating cell growth and differentiation or the polarized phenotype of cells. The major findings reported in this manuscript are summarized as follows:
1. HMT-3522 S1 cells express Cx43 and undergo acinar morphogenesis in 3-D cultures whereas S1 cells from which Cx43 has been knocked down fail to undergo acinar morphogenesis. This loss is accompanied by loss of apicolateral polarity in 3-D.
2. Knock-down of Cx43 causes mislocalization of β-catenin and Scrib from the apicolateral membrane domains.
3. Knockdown of Cx43 accelerated cell cycle entry, enhanced proliferation, and increased the cyclin D1 and c-Myc expression levels in 2-D and 3-D cultures without activating canonical Wnt/β-catenin signaling pathway.
4. Knockdown of Cx43 also activated ERK1/2 pathway as well as Rho GTPase signaling, a downstream effector of non-canonical Wnt/β-catenin signaling pathway.
5. Knockdown of Cx43 altered invasion only under conditions where stiffness of the extracellular matrix was reduced.
Based on these salient findings the authors concluded that Cx43 expression regulates the proliferation, invasion, and apicolateral polarity by altering non-canonical Wnt/β-catenin signaling pathway.
The major strength of this study is the choice of a well-characterized cell line that undergoes acinar morphogenesis, and which is well accepted to be an in vivo correlate of the normal human mammary epithelial cells. The authors have published several interesting papers related to the role of Cxs in regulating the behavior of mammary epithelial cells.
A major weakness of the proposed studies is the lack of link between alteration in cell proliferation, loss of apicolateral polarity, upregulation of cyclin D1 and c-Myc, activation of non-canonical Wnt/β-catenin signaling pathway and alteration in invasion under specific conditions. It is unclear to this reviewer as to how would knockdown of Cx43 induce all these changes! This has not been made clear in the discussion. In addition, there is no other control for the knockdown. The transformed counterpart of S1 cells, HMT-3522 T4-2 cells, were used only in one experiment. Another major pitfall of this study is referral to the data from a manuscript that has been submitted for publication and from which no Figures have been included for the reviewer to have a look. It appears to this reviewer that these data are crucial for the appreciation of this important study. The discussion is very long and superficial with no relevance to the key data whose possible interpretation is being discussed.
We thank the reviewer for reviewing the manuscript. We have worked on shortening and focusing the discussion, as per the reviewer's specific comments below. We now discuss the link between the loss of apical polarity, the enhanced proliferation and migration, the upregulation of c-Myc and cyclin D1 and the activation of noncanonical Wnt signaling.
The study we refer to as "submitted" (Bazzoun/Adissu et al.) has been under consideration for publication to J. Cell Science (JCS) well before finalizing the current study (by Fostok et al.) for Cancers. Our earlier JCS manuscript is currently under revision for resubmission by March 1, 2019. We have now attached, as per the reviewer's request, the initial submission to JCS inclusive of the figures that we refer to in our present submitted manuscript to Cancers.
The transfection and infection protocols are now elaborated on in the "Materials and Methods" section (page 18, lines 537-550). In our previously JCS submitted manuscript (Bazzoun/Adissu et al.) that is currently due for resubmission by March 1, 2019, we have transfected S1 cells with an empty vector (EV) control, nonsilencing sequence (NSS)-shRNA and Cx43-shRNA. NSS-shRNA S1 cells phenotypically resembled control S1 cells in 3-D cultures and did not display any alterations in their differentiation potential (see Figure 1E, top left panel for control S1 cells and Figure 3D, top left panel for NSS-shRNA S1 cells in the attached JCS submitted manuscript). In addition, control S1 cells and NSS-shRNA counterparts expressed similar mRNA and protein levels of Cx43 (see Figure 3A in the attached JCS submitted manuscript) and displayed apicolateral localization of Cx43 in acini (see Figure 1E, top left panel for control S1 cells and Figure 3D, top left panel for NSS-shRNA S1 cells in the attached JCS submitted manuscript). This is also now referred to in the Cancers manuscript under the "Discussion" section (page 15, lines 391-393). We have overexpressed Cx43 in the tight junction- and apical polarity-deficient MCF-10A cells (see Figure 4A, right panel in the attached JCS submitted manuscript), and we found that only MCF-10A acini with apical localization of Cx43 showed apical polarity (the apical localization of ZO-1 served as a marker for apical polarity). In contrast, MCF-10A acini overexpressing Cx43 non-apically failed to establish apical polarity (see Figure 4C in the attached JCS submitted manuscript). This suggests the importance of Cx43 localization in controlling the establishment of apical polarity in the mammary epithelium and rules out possible off-target effects. This is also now referred to in the Cancers manuscript under the "Discussion" section (page 15, lines 393-395). We are now attaching the JCS submitted manuscript with figures for the kind attention of the reviewer.
We have attended to the reviewer's comments and addressed all the specific points below.
The specific comments are as follows:
Introduction:
Line 52: The word “gap junctions” should replace the word channel, as it has not been mentioned that gap junctions are formed of cell-cell channels.
The channel-forming ability of gap junctions is now mentioned in the "Introduction" section (page 1, line 38). As per the reviewer's suggestion, we have now replaced the term "channel" with the term "gap junction" in the "Introduction" section (page 2, line 53). We have also consistently used the term "gap junction" instead of "channel" throughout the manuscript (page 6, line 156 and page 15, line 422).
Lines 56-57: It is a paradoxical statement. This sentence clearly shows that Cx expression affects Wnt signaling. So this sentence should be modified.
This statement is modified now (page 2, lines 57-59).
Lines 58-61: This is not a correct statement. Dissemination is linked to loss of cadherin expression but not to Cx expression. Please site the relevant reference or modify this statement.
This statement is modified now, and the relevant references (below) have been cited (page 2, lines 59-63).
Banerjee, D. Connexin’s connection in breast cancer growth and progression. International journal of cell biology 2016, 2016.
Carystinos, G.D.; Bier, A.; Batist, G. The role of connexin-mediated cell–cell communication in breast cancer metastasis. Journal of mammary gland biology and neoplasia 2001, 6, 431-440.
Results:
Lines 101-102: These are crucial data and should have been included in this manuscript.
The data on the loss of apical polarity and the lumen-forming ability in Cx43-shRNA S1 cells is part of an earlier manuscript (Bazzoun/Adissu et al.) that has been submitted to JCS before preparing the current manuscript (by Fostok et al.). Our earlier JCS manuscript is currently under revision for resubmission by March 1, 2019. We have attached the JCS manuscript with Figures 3E and 5D pertaining to that data for the reviewer to check. The loss of lumen formation in Cx43-shRNA S1 cells has been reproduced in the current study by Fostok et al. and appears in our Cancers manuscript (page 4, Figure 1B, left lower panel). This data is now referred to in the "Results" section under "Silencing Cx43 Triggers Cell Cycle Entry and Enhances the Proliferation Rate" (page 3, line 111).
Figure 1 B: Please describe clearly in the method how acinar sizes were measured and how many acini were scored? Measuring areas could be misleading when measuring proliferation in 3-D cultures.
This is now described clearly in the "Methods" section under "Trypan Blue Exclusion Method" (page 18, lines 555-560) and in the legend of Figure 1 (page 4, lines 130-132).
Figure 2: Expression level of Cx43 in S1 and knocked-down cells should by shown? Also, please describe the specifics of ShRNAs and the details of the vector used in “Methods”. How many sequences were tried and how stable was the knockdown?
The expression levels of Cx43 in S1 and Cx43-shRNA S1 cells is shown in Figure 4A (page 7). The transfection and infection protocols are now elaborated on in the "Materials and Methods" section (page 18, lines 537-550). The stability of the knockdown was regularly assessed in different cell passages by Western blot analysis throughout this study. This is now indicated in “Materials and Methods” section (page 18, lines 548-550).
Lines 146-148: This statement cannot be made with regard to work that is barely submitted. Please replace channel-independent with "gap junction-independent”.
We have attached the JCS submitted manuscript (Bazzoun/Adissu et al) with Figures 2D and 2G pertaining to this data for the reviewer’s convenience. We have also supported the results of our Cancers submitted work with previous findings from our lab that rule out the involvement of GJIC in the antiproliferative role of Cx43 (page 6, lines 153-155).
As stated above in response to the first “specific comment” by this reviewer, we have now replaced the term "channel" with the term "gap junction" throughout the manuscript (page 2, line 53, page 6, line 156 and page 15, line 422).
Lines 157-159: These data should be shown or this statement removed.
The statement pertaining to Cx43 localization in 3-D cultures of S1 cells has been removed (page 6, lines 167, 168), since this data is part of our JCS submitted work by Bazzoun/Adissu et al. (see Figure 1E, top left panel in the attached JCS submitted manuscript). This JCS manuscript is now attached for the reviewer’s convenience.
Figure 3: This Figure and legend is confusing. Also the Figure is split. In addition, the legend is hard to comprehend with regard to reference points in the Figure. Figure 3 legend does not describe Figure 3B. In addition, in 3-D cultures, β-catenin appears at the apical domain, which is unlikely. It should be on the lateral domain. Markers specific for the apical or basolateral domain would improve this Figure. Upper immunofluorescence panels show cells under different magnification or else knockdown of Cx43 increases cell size as well. Please clarify.
The figure legend has been modified (page 8, lines 200-211) for an overall better comprehension and for clearer description of part B.
β-catenin localizes at apicolateral membrane domains (apical side of the lateral membrane) in 3-D cultures of S1 cells. The signal observed in the center of the acinus originates from upper or lower optical sections of the acinus. In our previous JCS submitted work (Bazzoun/Adissu et al.), we have investigated the localization of apical and basal membrane markers (ZO-1 and integrin α-6, respectively), and we presented evidence for the establishment of apicobasal polarity in 3-D cultures of S1 cells on the basis of localization of those markers. We have attached the JCS submitted manuscript with Figure 2E showing this data for the reviewer’s convenience.
The upper immunofluorescence images were taken at the same magnification. In this study, we show that silencing Cx43 in S1 cells upregulates the activities of Cdc42 and Rac1, which are both involved in regulating actin polymerization, leading to an increase in protrusion [1]. Thus, the increase in area could be due to an exaggerated protrusion in Cx43-shRNA S1 cells, due to the increase in Cdc42 and Rac1 activities.
1. El-Sibai, M.; Nalbant, P.; Pang, H.; Flinn, R.J.; Sarmiento, C.; Macaluso, F.; Cammer, M.; Condeelis, J.S.; Hahn, K.M.; Backer, J.M. Cdc42 is required for EGF-stimulated protrusion and motility in MTLn3 carcinoma cells. Journal of cell science 2007, 120, 3465-3474.
Figure 4 legend line 195: Between 100-XXX acini were scored per group is a better statement.
We have modified the original statement (page 8, lines 203, 204, 211), as per the reviewer's suggestions.
Line 200: It is not clear from Figure 4 B, what has happened to the localization of β-catenin. In addition, the use of the word apicolateral domain is confusing. It should be clarified what the authors mean by apicolateral domain!
While β-catenin localizes at apicolateral membrane domains in 3-D cultures of S1 cells, it becomes mislocalized in Cx43-shRNA acini and loses this characteristic distribution pattern, as it additionally appears at basal membranes.
The term "apicolateral" is now defined in the "Introduction" section (page 2, line 77). Please refer also to the reply we provided for Figure 3 above.
Lines 220-222: Colocalization of Cx43 and β-catenin should be shown. This will help the readers appreciate the colocalization status of Cx43 and beta-catenin.
We have shown colocalization of Cx43 and β-catenin in our JCS submitted manuscript by Bazzoun/Adissu et al. Please refer to JCS Figure 1E. This JCS manuscript is due for resubmission with revisions by March 1, 2019. It is now included with this rebuttal for the reviewer's convenience. We refer to this data now in the Cancers manuscript (page 6, lines 166, 167).
Lines 264-266: It is not clear what the authors mean by the "phenotype".
We have modified the term "phenotype" into "invasive phenotype" to clarify what we mean (page 11, line 278).
Lines 266-268: This is a vague statement. Do the authors mean that the ECM stiffness increases invasiveness of breast tumors? Alternatively, do they mean something else? Please clarify.
We have modified this statement in order to clarify that increased ECM stiffness enhances invasion of premalignant and malignant mammary epithelial cells (page 10, line 277 and page 11, lines 278-282).
Line 271: The term "permissive ECM stiffness" should be clarified.
This term is now clarified in the "Introduction" section (page 2, lines 91, 92) and in the "Results" section (page 11, line 285 and page 14, line 355). In addition, the term "permissiveness" on page 17, line 519 is now replaced with "compliance".
Figure 6 B: There is a huge cell size difference between control and Cx43 knockdown cells in immunofluorescence images. It appears that we are dealing here with multiple population of cells.
The efficiency of Cx43 knockdown in our model is not 100%. The cells where Cx43 is knocked down have exaggerated actin-rich invadopodia-like structures (Figure 6B, lower panel; page 12) and seem larger. As mentioned in response to a previous comment of the reviewer, we show in this study that silencing Cx43 in S1 cells upregulates the activities of Cdc42 and Rac1, which are both involved in regulating actin polymerization, leading to an increase in protrusion [1]. Thus, the increase in area could be due to an exaggerated protrusion in Cx43-shRNA S1 cells, due to the increase in Cdc42 and Rac1 activities.
El-Sibai, M.; Nalbant, P.; Pang, H.; Flinn, R.J.; Sarmiento, C.; Macaluso, F.; Cammer, M.; Condeelis, J.S.; Hahn, K.M.; Backer, J.M. Cdc42 is required for EGF-stimulated protrusion and motility in MTLn3 carcinoma cells. Journal of cell science 2007, 120, 3465-3474.
Figure 7 legend: Apicolateral is a confusing term. Is Scrib localized apically or basolaterally or at the junction of apical and lateral domains? It should be clarified.
Scrib is localized at the apical side of the lateral membrane, i.e. at the apicolateral membrane domains. We have defined the term "apicolateral" in the "Introduction" section (page 2, line 77), as per an earlier comment on the localization of β-catenin. Please see also above rebuttal for Figures 3 and 4B.
Line 330 and paragraph: What is the driving force here? Is it activation of ERK1/2 pathway or Rho GTPase signaling? Is there a potential link between these pathways? A clear rationale would help the readers.
The rationale is clearly explained now in the "Results" section under "Silencing Cx43 Activates Rho GTPase Signaling" (page 14, lines 347-351). We have also mentioned the cross-talk between ERK1/2 and Rho GTPase signaling in the "Discussion" section (page 17, lines 491-497).
Figure 8: It looks like that all GTPases (Rho, Rac and Cdc42) are activated! Different GTPases do different things. A brief description or explanation would help the readers and would clarify what the authors are trying to emphasize specifically?
An explanation of why all three Rho GTPases are activated is now included in the "Discussion" section (page 17, lines 481-484).
Lines 349-370: These lines have nothing to do with the discussion. Please discuss the data and do not review the literature here. This is basically a repetition of the "Introduction" to some extent.
We have deleted the lines 370-374 (page 14) and 375-387 (page 15) to avoid redundancy with the introduction or review of literature.
Lines 372-373: Once again this is a very distracting statement. These data should be available if this reviewer is to provide constructive critique to the authors.
We have attached the JCS manuscript with the figures for the reviewer to check. Please refer to Figures 3E and 5D of the JCS submitted manuscript.
Lines 394-395: Again these data should be made available. Otherwise, it is not possible to appreciate the significance of this study.
As stated above, we now have attached the JCS manuscript with the figures for the reviewer to check. Please refer to Figures 2D and 2G of the JCS submitted manuscript.
Lines 400-403: This statement is not justified by the data shown in this manuscript. It should be modified or deleted.
We have deleted this statement (page 15, lines 424, 425).
Lines 418-427: These lines center on β-catenin with no reference to Cx43. These lines are diffuse and are not related to the main findings of the manuscript.
Lines 440-449 (page 16) have been deleted.
Lines 435-437: Please discuss the significance of upregulation of c-Myc and cyclin D1 with regard to Cx43 knockdown. This is potentially an important finding.
This is now tackled in the "Discussion" section (page 16, lines 452-459), as per the reviewer's suggestion.
METHODS
Lines 498-499: The details of which sequence was used to knock down Cx43 should be described. This key methodology is once again referred to as manuscript submitted is also discomforting.
The transfection and infection protocols are now elaborated on in the "Materials and Methods" section (page 18, lines 537-550).
Lines 506-508: It is not clear how exactly the acinar areas were measured and how many acini were scored.
This is now described clearly in the "Methods" section under "Trypan Blue Exclusion Method" (page 18, lines 555-560) and in the legend of Figure 1 (page 4, lines 130-132), as per an earlier comment of the reviewer.
Lines 520-521: Approximately how many acini were used and how many cells from these acini were subjected to cell cycle analysis by BD FACSArea III.
S1 and Cx43-shRNA S1 cells were seeded at an equal density in 3-D. Acini were isolated from the Matrigel and were then subjected to trypsin disaggregation. Ten thousand cells were analyzed by flow cytometry. This information is now included in the "Materials and Methods" section under "Cell Cycle Analysis" (page 19, lines 571-573).
Lines 593-598: How much total protein was used for the co-immunoprecipitation experiments.
This information is now included in the "Materials and Methods" section under "Co-Immunoprecipitation" (page 20, line 649).
This reviewer's comments to the manuscript written by Fostok et al. with regard to role of Cx43 in non-neoplastic breast epithelium were intended to help the authors. The authors have made substantial contributions with regard to the role of Cxs in regulating breast cancer cell proliferation and differentiation. However, an important set of references were omitted. Specifically, the authors failed to cite several important seminal findings reported by James Trosko's group (with CC Chang and others) which described the role of gap junctions in regulating the malignant phenotype, differentiation, and stem cell like characteristics of breast cancer epithelial cells both in 2-D and 3-D organotypic cultures. This reviewer thus encourages the authors to cite some of those references.
We have cited some of Dr. James Trosko's studies under the "Introduction" section (page 1, line 42).
I hope the authors would find these comments useful. Particularly, the discussion is diffuse and should be reduced to discussion of key points.
We would like to thank the reviewer for thoroughly revising our manuscript and for providing us with useful comments. We have worked on reducing and focusing the discussion, as per the reviewer's comments.
Reviewer 2 Report
In this manuscript, the authors used a nontumorigenic human mammary epithelial cell line HMT-3522 S1 cells to demonstrate roles of loss of Cx43 in triggering cell cycle entry and cellular invasion via the MAPK and non-canonical Wnt/Rho GTPase pathways. Overall the study is thorough and the manuscript is well written, although it can be improved by addressing the following points:
The entire study is based on shRNA-mediated knockdown of Cx43; it is unclear whether at least two shRNAs for Cx43 were used in the study, to rule out any off-target effect. Alternatively, a rescue experiment (e.g., by putting Cx43 back to the knockdown cells) may rule out any non-specific effect from the knockdown system.
The authors demonstrated that silencing Cx43 did not activate the canonical Wnt/beta-catenin pathway, but instead activated the Rho GTPase non-canonical Wnt signaling. Would the authors be able to reproduce some of the phenotype (e.g., increased motility and invasiveness upon Cx43-loss) by treating the wildtype cells with Rho GTPase inhibitors? Also how loss of Cx43 leads to activation of the GTPase signaling? The authors may want to at least discuss this key finding.
Author Response
Please note that referees’ comments appear below in bold. Rebut by the author appears in regular non-bold font.
Thank you
Comments and Suggestions for Authors
In this manuscript, the authors used a nontumorigenic human mammary epithelial cell line HMT-3522 S1 cells to demonstrate roles of loss of Cx43 in triggering cell cycle entry and cellular invasion via the MAPK and non-canonical Wnt/Rho GTPase pathways. Overall the study is thorough and the manuscript is well written, although it can be improved by addressing the following points:
We thank the reviewer for reviewing the manuscript. We have attended to the reviewer's comments and addressed all the points.
The entire study is based on shRNA-mediated knockdown of Cx43; it is unclear whether at least two shRNAs for Cx43 were used in the study, to rule out any off-target effect. Alternatively, a rescue experiment (e.g., by putting Cx43 back to the knockdown cells) may rule out any non-specific effect from the knockdown system.
The transfection and infection protocols are now elaborated on in the "Materials and Methods" section (page 18, lines 537-550). In our previously J. Cell Science (JCS) submitted manuscript (Bazzoun/Adissu et al.) that is currently due for resubmission by March 1, 2019, we have transfected S1 cells with an empty vector (EV) control, nonsilencing sequence (NSS)-shRNA and Cx43-shRNA. NSS-shRNA S1 cells phenotypically resembled control S1 cells in 3-D cultures and did not display any alterations in their differentiation potential (see Figure 1E, top left panel for control S1 cells and Figure 3D, top left panel for NSS-shRNA S1 cells in the attached JCS submitted manuscript). In addition, control S1 cells and NSS-shRNA counterparts expressed similar mRNA and protein levels of Cx43 (see Figure 3A in the attached JCS submitted manuscript) and displayed apicolateral localization of Cx43 in acini (see Figure 1E, top left panel for control S1 cells and Figure 3D, top left panel for NSS-shRNA S1 cells in the attached JCS submitted manuscript). This is also now referred to in the Cancers manuscript under the "Discussion" section (page 15, lines 391-393). We have overexpressed Cx43 in the tight junction- and apical polarity-deficient MCF-10A cells (see Figure 4A, right panel in the attached JCS submitted manuscript), and we found that only MCF-10A acini with apical localization of Cx43 showed apical polarity (the apical localization of ZO-1 served as a marker for apical polarity). In contrast, MCF-10A acini overexpressing Cx43 non-apically failed to establish apical polarity (see Figure 4C in the attached JCS submitted manuscript). This suggests the importance of Cx43 localization in controlling the establishment of apical polarity in the mammary epithelium and rules out possible off-target effects. This is also now referred to in the Cancers manuscript under the "Discussion" section (page 15, lines 393-395). We are now attaching the JCS submitted manuscript with figures for the kind attention of the reviewer.
The authors demonstrated that silencing Cx43 did not activate the canonical Wnt/beta-catenin pathway, but instead activated the Rho GTPase non-canonical Wnt signaling. Would the authors be able to reproduce some of the phenotype (e.g., increased motility and invasiveness upon Cx43-loss) by treating the wildtype cells with Rho GTPase inhibitors? Also how loss of Cx43 leads to activation of the GTPase signaling? The authors may want to at least discuss this key finding.
When wildtype cells are treated with inhibitors or siRNA of Rho GTPases, this leads to the inhibition of cell migration. We and others have previously shown that Rho GTPases (RhoA, Rac1 and Cdc42) are needed for breast cancer migration, due to their effect on actin polymerization [1-3] (below references). In this study, knocking down Cx43 in S1 cells leads to an increase in the migration and invasion phenotype, which correlates with an increase in the activation of Rho GTPases. This is consistent with the established role of RhoA, Rac1 and Cdc42 in the migratory and invasive phenotype of cancer cells. We have in fact previously shown that, compared to invasive breast cancer cells, the less invasive parental cells have a lower RhoA activation, as measured by FRET [1]. This is now elaborated on in the "Discussion" section (page 17, lines 481-483).
The knockdown of Cx43 has been shown to increase the activities of RhoA and Rac1 in fibroblasts [4], concomitantly with the enhanced migration. This study is mentioned in the "Discussion" section (page 17, lines 484-486), where we also propose now a mechanism through which the loss of Cx43 could activate Rho GTPase signaling in our model [5,6] (page 17, lines 486-491).
1. El-Sibai, M.; Pertz, O.; Pang, H.; Yip, S.-C.; Lorenz, M.; Symons, M.; Condeelis, J.S.; Hahn, K.M.; Backer, J.M. RhoA/ROCK-mediated switching between Cdc42- and Rac1-dependent protrusion in MTLn3 carcinoma cells. Exp Cell Res 2008, 314, 1540-1552, doi:10.1016/j.yexcr.2008.01.016.
2. El-Sibai, M.; Nalbant, P.; Pang, H.; Flinn, R.J.; Sarmiento, C.; Macaluso, F.; Cammer, M.; Condeelis, J.S.; Hahn, K.M.; Backer, J.M. Cdc42 is required for EGF-stimulated protrusion and motility in MTLn3 carcinoma cells. J Cell Sci 2007, 120, 3465-3474, doi:10.1242/jcs.005942.
3. Hanna, S.; Khalil, B.; Nasrallah, A.; Saykali, B.A.; Sobh, R.; Nasser, S.; El-Sibai, M. StarD13 is a tumor suppressor in breast cancer that regulates cell motility and invasion. Int J Oncol 2014, 44, 1499-1511, doi:10.3892/ijo.2014.2330.
4. Mendoza-Naranjo, A.; Cormie, P.; Serrano, A.E.; Hu, R.; O'Neill, S.; Wang, C.M.; Thrasivoulou, C.; Power, K.T.; White, A.; Serena, T. Targeting Cx43 and N-cadherin, which are abnormally upregulated in venous leg ulcers, influences migration, adhesion and activation of Rho GTPases. PloS one 2012, 7, e37374.
5. Asnaghi, L.; Vass, W.; Quadri, R.; Day, P.; Qian, X.; Braverman, R.; Papageorge, A.; Lowy, D. E-cadherin negatively regulates neoplastic growth in non-small cell lung cancer: role of Rho GTPases. Oncogene 2010, 29, 2760.
6. Zebda, N.; Tian, Y.; Tian, X.; Gawlak, G.; Higginbotham, K.; Reynolds, A.B.; Birukova, A.A.; Birukov, K.G. Interaction of p190RhoGAP with C-terminal domain of p120-catenin modulates endothelial cytoskeleton and permeability. Journal of Biological Chemistry 2013, jbc. M112. 432757.
Round 2
Reviewer 1 Report
The authors have responded to my critique adequately. It is nice to know that they found this reviewer's comments helpful. These changes will also expand the significance of their findings to general readers outside the field of gap junctions.